# Contrastive Spectral Rectification: Test-Time Defense towards Zero-shot Adversarial Robustness of CLIP

**Sen Nie** [1 2]  **Jie Zhang** [1 2]  **Zhuo Wang** [3]  **Shiguang Shan** [1 2]  **Xilin Chen** [1 2]

## Abstract

Vision-language models (VLMs) such as CLIP have demonstrated remarkable zero-shot generalization, yet remain highly vulnerable to adversarial examples (AEs). While test-time defenses are promising, existing methods fail to provide sufficient robustness against strong attacks and are often hampered by high inference latency and task-specific applicability. To address these limitations, we start by investigating the intrinsic properties of AEs, which reveals that AEs exhibit severe feature inconsistency under progressive frequency attenuation. We further attribute this to the model's inherent spectral bias. Leveraging this insight, we propose an efficient test-time defense named **C**ontrastive **S**pectral **R**ectification (CSR). CSR optimizes a rectification perturbation to realign the input with the natural manifold under a spectral-guided contrastive objective, which is applied input-adaptively. Extensive experiments across 16 classification benchmarks demonstrate that CSR outperforms the SOTA by an average of **18.1%** against strong APGD with modest inference overhead. Furthermore, CSR exhibits broad applicability across diverse visual tasks. Code is available at https://github.com/Summu77/CSR.

## 1 Introduction

Large-scale pre-trained Vision-Language Models (VLMs), notably CLIP (Radford et al., 2021), have revolutionized multimodal representation learning with their remarkable zero-shot generalization (Shen et al., 2022; Zhang et al., 2024a; Awais et al., 2025). Despite this success, CLIP models remain highly vulnerable to adversarial examples (Cui et al., 2024; Zhang et al., 2025a). Imperceptible perturba-

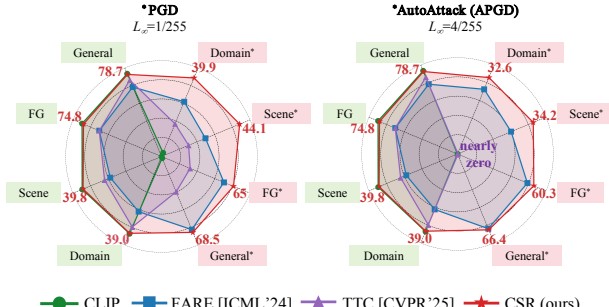

*Figure 1.* Zero-shot adversarial robustness comparison. We evaluate our CSR against CLIP, FARE (adversarial fine-tuning), and TTC (test-time defense) on 16 datasets grouped into General, Fine-Grained (FG), Scene, and Domain. The radar charts show Top-1 accuracy on benign and adversarial samples under standard PGD ($\epsilon = 1/255$) and the stronger AutoAttack (APGD) ($\epsilon = 4/255$).

tions can deceive the model, undermining its reliability in safety-critical open-world applications (Chen et al., 2025a).

To mitigate this vulnerability, Adversarial Fine-tuning (AFT) (Mao et al., 2023; Wang et al., 2024b) is a straightforward defense strategy by fine-tuning the model on adversarial examples. However, AFT incurs prohibitive computational costs for large-scale models and often severely degrades the model's performance on benign samples, as illustrated by FARE (Schlarmann et al., 2024) in Figure 1. Consequently, test-time defense has emerged as a promising alternative (Zhang et al., 2025c; Tong et al., 2025), aiming to purify inputs during inference without retraining. However, current methods often suffer from high inference latency (Zhu et al., 2025) and exhibit sub-optimal performance especially under strong attacks. For instance, as shown in Figure 1, TTC (Xing et al., 2025) collapses under stronger AutoAttack (Croce & Hein, 2020), yielding marginal robustness gains. Moreover, many defenses are inherently designed for specific scenarios (Li et al., 2024b; Sheng et al., 2025; Wang et al., 2025a), limiting their applicability to broader visual tasks such as Segmentation, Image Captioning, and Visual Question Answering (VQA). Therefore, developing an effective, efficient, and universal test-time defense for CLIP remains a critical challenge.

To this end, we start by investigating the intrinsic properties of adversarial examples (AEs). Intuitively, subtle adversarial perturbations should be fragile; however, prior efforts apply-

---

[1] State Key Laboratory of AI Safety, Institute of Computing Technology, Chinese Academy of Sciences, China, Beijing [2] University of Chinese Academy of Sciences, China, Beijing [3] University of Science and Technology of China, China, Hefei. Correspondence to: Jie Zhang <zhangjie@ict.ac.cn>.

*Proceedings of the 43rd International Conference on Machine Learning*, Seoul, South Korea. PMLR 306, 2026. Copyright 2026 by the author(s).

ing random noise to adversarial examples (Xing et al., 2025) failed to substantiate this property. We thus turn to the frequency domain, analyzing feature consistency via progressive frequency attenuation. As shown in Figure 3, benign features degrade smoothly with the progressive removal of mid-to-high frequencies, whereas adversarial features suffer a sharp decay, revealing a severe fragility across diverse attacks. We trace this phenomenon to a two-fold vulnerability intrinsic to CLIP: its spectral bias toward mid-to-high frequencies and its hypersensitivity to perturbations within these bands. Consequently, adversarial attacks inevitably exploit these high-impact spectral regions, rendering AEs inherently fragile to such spectral filtering.

Motivated by this insight, we propose **C**ontrastive **S**pectral **R**ectification (CSR), an efficient test-time defense method. Unlike the naive adoption of low-pass filtered features, CSR introduces an active mechanism that optimizes a learnable rectification perturbation superimposed onto the input image. Specifically, we formulate a contrastive objective where the low-pass filtered feature serves as a positive anchor and the original feature as a negative anchor. This objective drives the rectification perturbation to repel the image from the adversarial subspace and realign it with the natural manifold, as depicted in Figure 2. To ensure efficiency and maintain performance on benign examples, we incorporate an input-adaptive gating mechanism: by measuring feature consistency before and after spectral filtering, CSR triggers the rectification process only for samples exhibiting adversarial instability. Consequently, CSR leverages the intrinsic spectral properties of CLIP to efficiently detect and rectify adversarial examples at inference without retraining.

Through extensive experiments across 16 zero-shot classification benchmarks, we demonstrate that CSR consistently outperforms state-of-the-art defenses. Specifically, it achieves an average improvement of 6.9% against PGD and a substantial **18.1%** gain against the stronger AutoAttack (APGD), while maintaining accuracy on benign samples. Notably, CSR delivers this superior robustness with modest inference overhead, offering a highly efficient test-time solution. Furthermore, CSR generalizes effectively to diverse visual tasks, including Semantic Segmentation, Image Captioning, and VQA, highlighting its broad applicability.

Our contributions are as follows:

- We uncover a spectral fragility in adversarial examples and trace this vulnerability to CLIP's spectral bias and sensitivity to mid-to-high frequency components.
- We introduce CSR, a test-time defense that utilizes a spectral-guided contrastive rectification and an input-adaptive gating mechanism to efficiently detect and purify adversarial examples during inference.
- We validate CSR across 16 benchmarks, where it delivers state-of-the-art robustness (*e.g.*, +18.1% against APGD)

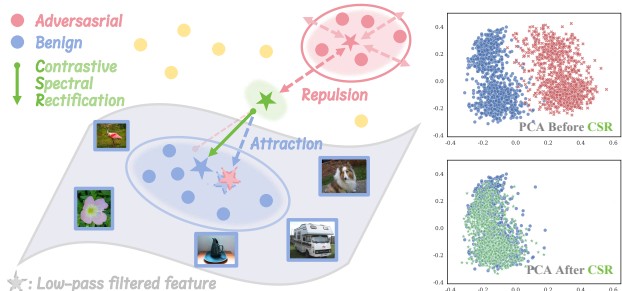

*Figure 2.* Mechanism of Contrastive Spectral Rectification (CSR). Leveraging a spectral contrastive strategy, CSR exerts **repulsion** from the original adversarial feature (solid red star) within the *adversarial subspace*, while inducing **attraction** toward the low-pass filtered feature (dashed red star)—an approximation on the *benign manifold*. This synergy steers the optimization toward the ground-truth feature (solid blue star), effectively rectifying the feature space as corroborated by the PCA visualization (right) on 300 images.

with high inference efficiency. Beyond classification, CSR also proves effective in diverse tasks.

## 2 Preliminaries and Related Work

Large pre-trained Vision-Language Models (VLMs), notably CLIP (Radford et al., 2021), learn aligned multimodal representations via contrastive learning, enabling remarkable zero-shot generalization across various tasks.

**Zero-shot Classification with CLIP.** Representatively, CLIP can leverage its cross-modal alignment to categorize images into open-vocabulary classes without task-specific fine-tuning. Formally, let $f(\cdot)$ and $g(\cdot)$ denote the image and text encoders, respectively. Given an input image $\boldsymbol{x}$ and a set of $K$ candidate classes, we construct textual prompts $\{\boldsymbol{t}_i\}_{i=1}^K$ using a template (*e.g.*, "a photo of a {class}"). Let $\boldsymbol{z}_v = f(\boldsymbol{x})/\|f(\boldsymbol{x})\|_2$ and $\boldsymbol{z}_{t,i} = g(\boldsymbol{t}_i)/\|g(\boldsymbol{t}_i)\|_2$ be the $L_2$-normalized visual and textual embeddings. The zero-shot prediction probability for the $i$-th class is computed via the softmax-normalized cosine similarities:

$$p(y_i|\boldsymbol{x}) = \frac{\exp(\tau \cdot \boldsymbol{z}_v^\top \boldsymbol{z}_{t,i})}{\sum_{j=1}^K \exp(\tau \cdot \boldsymbol{z}_v^\top \boldsymbol{z}_{t,j})}, \qquad (1)$$

where $\tau$ is a temperature parameter for scaling, and the predicted label is given by $\hat{y} = \arg\max_i p(y_i|\boldsymbol{x})$.

**Adversarial Attacks on CLIP.** Despite its impressive generalization, CLIP remains susceptible to adversarial perturbations. Formally, attackers seek an imperceptible perturbation $\boldsymbol{\delta}$ within an $\ell_p$-norm budget $\epsilon$ to maximize the loss $\mathcal{L}$, typically the cross-entropy loss for classification:

$$\max_{\boldsymbol{\delta}} \mathcal{L}(f(\boldsymbol{x} + \boldsymbol{\delta}), g(\boldsymbol{t}_y)) \quad \text{s.t.} \quad \|\boldsymbol{\delta}\|_p \le \epsilon. \qquad (2)$$

Standard methods, such as PGD (Madry et al., 2017) and the stronger AutoAttack (Croce & Hein, 2020), employ iterative gradient ascent to optimize this perturbation by lever-

aging the loss gradient ($\nabla_x \mathcal{L}$). A more extensive review of adversarial attacks on (L)VLMs is deferred to Appendix A.

To mitigate these threats, various defenses have emerged, generally falling into adversarial training and test-time defense.

**Adversarial Training for VLMs.** Early strategies, termed Adversarial Fine-Tuning, enhance robustness by fine-tuning models on adversarial examples. Approaches like TeCoA (Mao et al., 2023) and PMG-AFT (Wang et al., 2024b) introduce auxiliary objectives to mitigate catastrophic forgetting (Wang et al., 2024c; Dong et al., 2025b; Sui et al., 2025; Dong et al., 2025c), while FARE (Schlarmann et al., 2024) and Sim-CLIP+ (Hossain & Imteaj, 2024) employ unsupervised learning to improve robustness transferability. However, the prohibitive cost of full fine-tuning has shifted focus toward Adversarial Prompt Tuning, leveraging parameter-efficient techniques (Ding et al., 2023; Han et al., 2024; Yuan et al., 2025). By optimizing learnable prompts while keeping the backbone frozen, methods such as VPT (Mao et al., 2023), APT (Li et al., 2024a), FedAPT (Zhai et al., 2025), and AdvPT (Zhang et al., 2024b) effectively align text embeddings with adversarial visual features. Subsequent works (*e.g.*, FAP (Zhou et al., 2024b), CoAPT (Wang et al., 2025c)) further advance this paradigm by incorporating few-shot regimes and structural priors. Despite these improvements, these methods frequently entail high training costs and risk compromising benign performance or overfitting.

**Test-Time Defense for VLMs.** Drawing inspiration from Test-Time Adaptation (Shu et al., 2022; Abdul Samadh et al., 2023; Dong et al., 2025a; Chen et al., 2025c; Zhou et al., 2025; Hu et al., 2025), recent research focuses on enhancing CLIP's robustness at inference without retraining. One direction focuses on textual alignment (Zhou et al., 2024a; Hussein et al., 2024): TAPT (Wang et al., 2025a), R-TPT (Sheng et al., 2025), and D-TPT (Han & Hwang, 2025) employ entropy-based prompt tuning, while COLA (Zhu et al., 2025) formulates the alignment as an optimal transport problem. Parallel efforts target visual purification (Deng et al., 2025; Su & Balogh, 2025): CLIPure (Zhang et al., 2025c) leverages diffusion priors, while TTE (Pérez et al., 2021), DDA (Reyes-Amezcua et al., 2024), and AOM (Tong et al., 2025) utilize transformation ensembles. Closest to our work is TTC (Xing et al., 2025), which updates inputs to counteract perturbations. However, its only focus on escaping the adversarial subspace without benign guidance distorts natural semantics, thereby limiting its efficacy. Overall, existing methods remain constrained by their vulnerability to strong attacks, high inference latency, and limited task-specific applicability, motivating our proposal of an effective, efficient, and universal test-time defense method.

Comparisons with prior frequency-robustness studies are deferred to Appendix A.

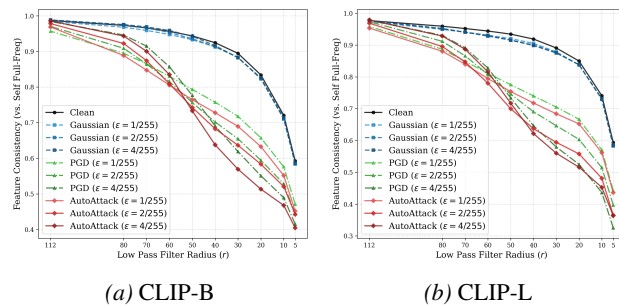

*(a)* CLIP-B       *(b)* CLIP-L

*Figure 3.* Spectral Consistency Disparity. Cosine similarity between original and low-pass filtered embeddings across decaying bandwidth radii $r$. While benign examples maintain high semantic fidelity, adversarial examples exhibit rapid feature collapse. See Appendix D for consistent trends across additional datasets.

## 3 Insight

### 3.1 Observation: Disparity in Feature Consistency

To motivate our test-time defense, we first investigate the intrinsic properties of adversarial examples (AEs). Specifically, we analyze the spectral consistency of CLIP by examining the stability of feature representations under low-pass filtering. Using 1,000 images sampled from ImageNet, we apply filters with decaying bandwidth radii $r$ to three categories: clean images, Gaussian-corrupted counterparts, and AEs generated via PGD (Madry et al., 2017) and AutoAttack (Croce & Hein, 2020). We quantify feature resilience using the cosine similarity between the filtered and original embeddings. As shown in Figure 3, a divergence emerges:

**(i) Stability of Benign Examples (BEs):** Benign Examples, encompassing both clean inputs and Gaussian-noised counterparts, exhibit high feature stability, maintaining semantic fidelity even under aggressive frequency truncation. Notably, samples corrupted with Gaussian noise show negligible feature divergence from the clean baseline, even at relatively high noise budgets of $\ell_\infty = 4/255$.

**(ii) Fragility of Adversarial Examples (AEs):** In contrast, AEs suffer from abrupt degradation in feature consistency as mid-to-high frequency components are removed. This vulnerability is pronounced even at minimal noise budgets (*e.g.*, $\ell_\infty = 1/255$) and persists across diverse attack variants. This indicates that adversarial deceptive signals are critically reliant on mid-to-high frequency components.

**Discussion.** We attribute the resilience of BEs to the robust semantic priors instilled by large-scale pre-training. This fosters a strong shape bias (Naseer et al., 2021), guiding the model to prioritize structural representations that reside firmly on the low-dimensional manifold of natural images. However, the divergent behavior of AEs prompts a fundamental question: *Why do adversarial perturbations inherently gravitate toward the mid-to-high frequency spectrum, and in what ways does this spectral bias bolster their deceptive efficacy?* We explore the underlying mechanisms

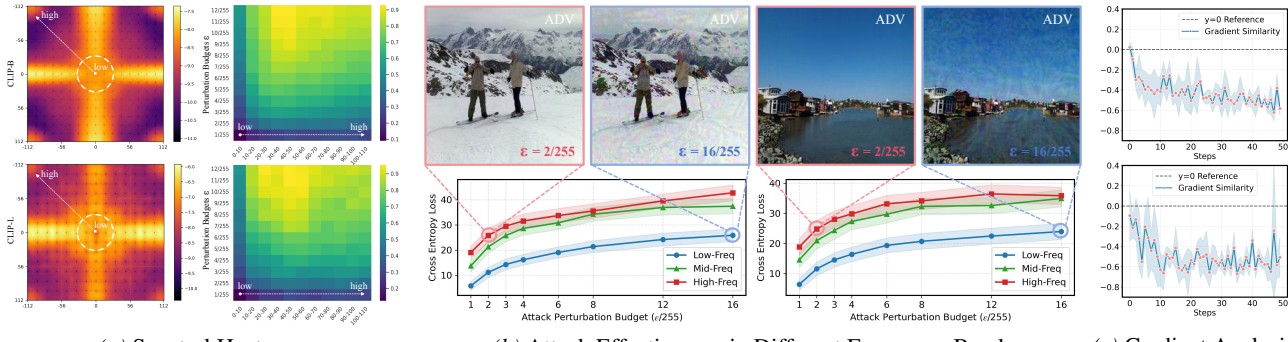

*(a)* Spectral Heatmaps       *(b)* Attack Effectiveness in Different Frequency Bands       *(c)* Gradient Analysis

*Figure 4.* Analysis. *(a)* The gradient magnitude (left) and representational shift (right) are concentrated in mid-to-high frequency components, indicating that the model is most vulnerable to perturbations in these bands. *(b)* Attacks constrained to low frequencies are inefficient. *(c)* The adversarial gradient exhibits consistent negative cosine similarity with the low-frequency constraint gradient.

of this frequency preference in the following subsection.

### 3.2 Analysis: Spectral Bias and Sensitivity of CLIP

Since AEs are crafted to maximize model error under constrained budgets, their consistent gravitation toward mid-to-high frequency components suggests a spectral vulnerability within CLIP. To deconstruct this, we examine the spectral density of loss gradients and quantify the representational shift induced by band-specific perturbations.

**(i) Intrinsic Spectral Bias:** We formally characterize the sensitivity of the model's objective function with respect to the input's frequency components. Let $\mathcal{F}(\boldsymbol{x}) \in \mathbb{C}^{H \times W}$ denote the 2D Discrete Fourier Transform of input $\boldsymbol{x}$. We define the **S**pectral **G**radient **M**agnitude (SGM), $\mathcal{S} \in \mathbb{R}^{H \times W}$, as the expected magnitude of the gradient of the loss $\mathcal{L}$ with respect to the spectral amplitude:

$$\mathcal{S}(u, v) = \mathbb{E}_{\boldsymbol{x} \sim \mathcal{D}} \left[ \left\| \nabla_{|\mathcal{F}(\boldsymbol{x})_{u,v}|} \mathcal{L}(f(\boldsymbol{x}), g(\boldsymbol{t}_y)) \right\|_2 \right], \quad (3)$$

where $(u, v)$ represents the spatial frequency coordinates. As visualized in Figure 4a (Left), the heatmap of $\mathcal{S}$ reveals a distinct vulnerability: CLIP exhibits significant gradient responses $\nabla \mathcal{L}$ extending well into the mid-to-high frequency periphery (the area outside the white dashed circle). This distribution uncovers a spectral bias: unlike human perception, which is robustly anchored in low-frequency semantics (*e.g.*, shape), CLIP places disproportionate predictive weight on mid-to-high frequency patterns. These components, identified as non-robust features (Ilyas et al., 2019), yield steep gradients and serve as an accessible "discriminative shortcut" for attackers to efficiently manipulate model predictions.

**(ii) Spectral Feature Hypersensitivity:** We quantify the sensitivity of CLIP's feature representations to band-specific spectral perturbations. We measure the worst-case feature space displacement induced by attacks restricted to specific frequency bands. Let $\mathbf{M}_k$ be a binary mask isolating the $k$-th frequency band. We solve for the optimal perturbation

$\boldsymbol{\delta}_k$ that maximizes the feature drift $\Delta_{\Phi}$ with PGD:

$$\max_{\boldsymbol{\delta}_k} \Delta_{\Phi} = 1 - \frac{f(\boldsymbol{x})^{\top} f(\boldsymbol{x} + \mathcal{F}^{-1}(\mathbf{M}_k \odot \mathcal{F}(\boldsymbol{\delta}_k)))}{\|f(\boldsymbol{x})\|_2 \|f(\boldsymbol{x} + \mathcal{F}^{-1}(\mathbf{M}_k \odot \mathcal{F}(\boldsymbol{\delta}_k)))\|_2}, \quad (4)$$

s.t. $\|\boldsymbol{\delta}_k\|_{\infty} \leq \epsilon.$

The results in Figure 4a (Right) reveal a sensitivity imbalance. While perturbations restricted to low-frequency bands (*e.g.*, 0-30) induce small feature shifts $\Delta_{\Phi}$, especially under low noise budgets ($\ell_{\infty} < 4/255$), those targeting mid-to-high frequency bands trigger large representational deviations. This indicates that the model's representation is hypersensitive to mid-to-high frequency components. Consequently, these bands constitute a structural vulnerability, allowing minimal noise to efficiently displace the image embedding from its natural semantic manifold.

Finally, we investigate the effectiveness of attacks across different frequency bands. Our analysis reveals an efficiency-imperceptibility trade-off, as shown in Figure 4b. Specifically, confining perturbations to the low-frequency band results in inefficiency, necessitating prohibitive budgets (*e.g.*, $\ell_{\infty} = 16/255$) to match the damage of high-frequency attacks (*e.g.*, $\ell_{\infty} = 2/255$). This magnitude amplification introduces conspicuous wavy artifacts, thereby violating the imperceptibility premise. We attribute this to an inherent gradient incompatibility. As evidenced in Figure 4c, the gradient for adversarial aggressiveness exhibits consistent negative cosine similarity with that of low-frequency confinement (theoretical analysis in Appendix C). Based on the above analysis, **adversarial attacks naturally gravitate toward the mid-to-high frequency bands to maximize efficiency**. Though shaped by training data, paradigms, and architectures (Maiya et al., 2021), this preference is consistent across large-scale pre-trained VLMs (see Appendix B).

## 4 Method

Building on the spectral insights derived in Section 3, we propose **C**ontrastive **S**pectral **R**ectification (CSR), a test-time defense framework designed to robustify CLIP against adversarial perturbations without retraining. The overall

**Algorithm 1** Contrastive Spectral Rectification (CSR)

**Require:** Input image $x$; CLIP image encoder $f(\cdot)$; Gaussian LPF $G_r(\cdot)$ with radius $r$; detection threshold $\tau$; perturbation budget $\epsilon$; rectification steps $N$; step size $\alpha$.

**Ensure:** Rectified reliable image $x^*$

1: $x_{low} \leftarrow G_r(x)$
2: $\mathcal{C}(x) \leftarrow \text{sim}(f(x), f(x_{low}))$      ▷ *(i) AEs Detection*
3: **if** $\mathcal{C}(x) \geq \tau$ **then**
4:     **return** $x$      ▷ *Consistent → Benign Image*
5: **else**
6:     $\delta \leftarrow 0$, $x' \leftarrow x + \delta$, $x^* \leftarrow x$, $\mathcal{L}_{best} \leftarrow -\infty$
7:     **for** $t = 1$ **to** $N$ **do**
8:        ▷ *(ii) Few-step Contrastive Spectral Rectification*
9:        $\mathcal{L}_{rec} \leftarrow \text{sim}(f(x'), f(x_{low})) - \text{sim}(f(x'), f(x))$
10:       **if** $\mathcal{L}_{rec} > \mathcal{L}_{best}$ **then**
11:              ▷ *(iii) Greedy Selection*
12:         $\mathcal{L}_{best} \leftarrow \mathcal{L}_{rec}$, $x^* \leftarrow x'$
13:       **end if**
14:       $\delta \leftarrow \text{clip}(\delta + \alpha \cdot \text{sign}(\nabla_{x'}\mathcal{L}_{rec}), -\epsilon, \epsilon)$
15:       $x' \leftarrow x + \delta$
16:     **end for**
17: **end if**
18: **return** $x^*$      ▷ *Return Rectified Reliable Image*

procedure is summarized in Algorithm 1.

### 4.1 Adversarial Detection via Spectral Consistency

The first step of CSR involves the input-adaptive gating mechanism. We employ a Gaussian low-pass filter $G_r(\cdot)$ with radius $r$ to attenuate mid-to-high frequency components, yielding a smoothed counterpart $x_{low} = G_r(x)$. As established in Section 3, benign samples exhibit strong feature consistency between their full-spectrum and low-frequency representations. Conversely, for adversarial examples, the removal of mid-to-high frequency components leads to a significant feature collapse. We quantify this divergence using the cosine similarity between the embeddings:

$$\mathcal{C}(x) = \text{sim}(f(x), f(x_{low})) = \frac{f(x)^\top f(x_{low})}{\|f(x)\|_2 \|f(x_{low})\|_2}. \quad (5)$$

A high $\mathcal{C}(x)$ indicates that the semantic content is robustly anchored in the low-frequency band, characteristic of BEs. Conversely, a low score reveals a reliance on fragile mid-to-high frequencies—a signature of AEs. We classify an input as adversarial if $\mathcal{C}(x) < \tau$, where $\tau$ is a detection threshold to balance detection sensitivity and the false positive rate. Detection ROC curves are provided in Appendix E.

### 4.2 Contrastive Spectral Rectification

While filtering is effective for detection, naive application of low-pass filters risks over-smoothing, eroding fine-grained details essential for zero-shot recognition. To overcome this,

we propose Contrastive Spectral Rectification. Instead of the passive suppression of mid-to-high frequency components, CSR formulates an active test-time optimization that reconstructs a reliable input $x^*$ from the adversarial query.

We freeze the model parameters and optimize a rectification perturbation $\delta$ to get the rectified sample $x' = x + \delta$. The objective function, $\mathcal{L}_{rec}$, is designed as follows:

$$\mathcal{L}_{rec}(\delta) = \underbrace{\text{sim}(f(x+\delta), f(x_{low}))}_{\text{Attraction Term}} - \lambda \cdot \underbrace{\text{sim}(f(x+\delta), f(x))}_{\text{Repulsion Term}}, \quad (6)$$

where $\text{sim}(\cdot, \cdot)$ denotes cosine similarity.

- **Attraction Term:** We utilize the low-frequency anchor $f(x_{low})$ as a benign semantic guide to pull the optimization trajectory back towards the robust natural manifold.

- **Repulsion Term:** To prevent stagnation in the adversarial state, this term pushes the sample away from the initial malicious adversarial embedding $f(x)$.

We solve Eq. (6) using Projected Gradient Descent (PGD) for a limited number of steps $N$. However, simply taking the final step's output is suboptimal, as the optimization trajectory on the non-convex manifold may oscillate. To ensure the reliability of the rectified sample, we adopt a **Greedy Selection** strategy. Specifically, during the optimization process, we monitor $\mathcal{L}_{rec}$ at each step $t$ and maintain the candidate $x^*$ that achieves the maximum objective value:

$$x^* = x + \underset{\delta_t \in \{\delta_1,...,\delta_N\}}{\arg\max} \mathcal{L}_{rec}(\delta_t). \quad (7)$$

This strategy ensures that the returned sample corresponds to the state of maximal spectral consistency and adversarial divergence, providing a stable input for the final inference.

## 5 Experiments

### 5.1 Setup

**Datasets and Models.** Following previous work, we employ a comprehensive benchmark suite of 16 datasets. Specifically, we evaluate on general object recognition (ImageNet (Deng et al., 2009), CIFAR-10/100 (Krizhevsky et al., 2009), STL10 (Coates et al., 2011), Caltech-101/256 (Fei-Fei et al., 2004; Griffin et al., 2007)), fine-grained classification (OxfordPets (Parkhi et al., 2012), Flowers102 (Nilsback & Zisserman, 2008), Food101 (Bossard et al., 2014), StanfordCars (Krause et al., 2013)), scene recognition (SUN397 (Xiao et al., 2010), Country211 (Radford et al., 2021)), and domain-specific applications (FGVCAircraft (Maji et al., 2013), EuroSAT (Helber et al., 2019), DTD (Cimpoi et al., 2014), PCAM (Veeling et al., 2018)). Furthermore, we conduct experiments on various tasks using the VOC2010 (Everingham et al., 2010) and MS-COCO (Lin et al., 2014) datasets. We adopt CLIP ViT-B/16

*Table 1.* Top-1 zero-shot classification accuracy (%) under 10-step PGD attacks ($\ell_\infty = 1/255$). "Clean" and "Rob." denote accuracies on benign and adversarial samples, respectively. The final two columns report the performance of our CSR compared to the original CLIP.

| Dataset | | Original | | Adversarial Fine-tuning | | | | Test-Time Defense | | | | | | | | | | | | | | | Δ | |
| | | CLIP | | TeCoA | | FARE | | R-TPT | | LPF | | HD | | Anti-Adv | | TTE | | TTC | | CSR (Ours) | | | |
| Type | Name | Clean | Rob. | Clean | Rob. | Clean | Rob. | Clean | Rob. | Clean | Rob. | Clean | Rob. | Clean | Rob. | Clean | Rob. | Clean | Rob. | Clean | Rob. | Clean | Rob. |
|---|---|---|---|---|---|---|---|---|---|---|---|---|---|---|---|---|---|---|---|---|---|---|---|
| General | ImageNet | 63.9 | 0.0 | 56.3 | 53.6 | 48.0 | 46.7 | **66.7** | 51.0 | 58.1 | 30.5 | 59.7 | 4.1 | 61.5 | 23.9 | 66.2 | 23.2 | 40.9 | 27.8 | 62.5 | **58.9** | -1.4 | +58.9 |
| | CIFAR10 | 88.1 | 0.5 | 63.7 | 63.8 | 61.4 | 61.2 | 81.6 | 69.2 | 89.0 | 40.4 | 84.1 | 11.8 | 82.8 | 63.5 | 85.5 | 29.8 | **90.0** | 28.2 | 87.2 | **75.0** | -0.9 | +74.5 |
| | CIFAR100 | 59.6 | 0.1 | 37.9 | 37.8 | 36.2 | 36.2 | 51.8 | 36.4 | 63.4 | 19.6 | 57.6 | 7.9 | 51.5 | 34.9 | 60.4 | 14.2 | **63.1** | 11.1 | 59.4 | **45.3** | -0.2 | +45.2 |
| | STL10 | 97.5 | 4.8 | 90.7 | 90.6 | 92.0 | 92.0 | 96.8 | **92.7** | 96.9 | 77.6 | 96.8 | 34.0 | 97.2 | 83.1 | **97.6** | 75.6 | 96.4 | 51.1 | 97.2 | 87.5 | -0.3 | +82.7 |
| | Caltech101 | 83.5 | 1.3 | 77.1 | 76.3 | 83.7 | **82.0** | 86.1 | 80.9 | 81.3 | 65.6 | 82.6 | 28.4 | 82.3 | 58.2 | **87.2** | 61.2 | 75.8 | 31.3 | 83.2 | 75.3 | -0.3 | +74.0 |
| | Caltech256 | 83.5 | 1.5 | 72.0 | 69.5 | 78.4 | 77.6 | **88.0** | 80.5 | 82.8 | 66.5 | 80.4 | 20.9 | 81.6 | 55.8 | 87.3 | 59.4 | 73.4 | 41.8 | 82.9 | 76.6 | -0.6 | +75.1 |
| Fine-Grained | OxfordPets | 88.9 | 0.0 | 71.3 | 69.3 | 79.3 | **74.3** | 85.8 | 68.3 | 79.4 | 41.7 | 84.2 | 4.0 | 86.5 | 36.7 | 84.8 | 12.5 | 78.8 | 27.0 | **88.9** | 65.9 | +0.0 | +65.9 |
| | Flowers102 | 66.0 | 0.0 | 21.3 | 21.5 | 41.7 | 38.7 | **65.8** | 48.8 | 59.8 | 33.1 | 63.3 | 3.5 | 63.5 | 25.3 | 65.3 | 5.5 | 55.8 | 23.3 | **65.8** | 53.5 | -0.2 | +53.5 |
| | Food101 | 84.8 | 0.0 | 30.6 | 29.2 | 47.0 | 44.5 | **86.6** | 68.9 | 79.6 | 38.6 | 86.0 | 1.1 | 83.9 | 29.3 | 85.3 | 24.8 | 57.5 | 33.2 | 81.9 | **80.7** | -2.9 | +80.7 |
| | StanfordCars | 65.2 | 0.0 | 32.9 | 23.6 | **70.7** | 67.9 | 68.6 | 45.4 | 54.0 | 16.9 | 57.8 | 1.4 | 62.5 | 13.7 | 59.0 | 14.4 | 46.6 | 20.2 | 62.7 | 60.9 | -2.5 | +60.9 |
| Scene | SUN397 | 63.6 | 0.2 | 39.9 | 37.2 | 47.6 | 45.7 | 64.1 | 53.1 | 59.2 | 31.3 | 59.7 | 4.0 | 62.5 | 22.7 | **65.4** | 20.2 | 47.5 | 26.2 | 63.0 | **66.3** | -0.6 | +66.1 |
| | Country211 | 17.0 | 0.0 | 3.0 | 2.7 | 4.4 | 3.9 | **19.2** | 8.9 | 14.7 | 2.5 | 15.1 | 0.0 | 15.2 | 1.9 | 15.8 | 0.3 | 10.2 | 4.4 | 16.6 | **22.9** | -0.4 | +22.9 |
| Domain | FGVCAircraft | 23.1 | 0.0 | 5.2 | 7.9 | 12.2 | 12.6 | **23.8** | 16.7 | 17.8 | 7.2 | 18.8 | 1.3 | 20.4 | 6.0 | 23.3 | 4.9 | 13.8 | 10.6 | 22.7 | **31.2** | -0.4 | +31.2 |
| | EuroSAT | 42.9 | 0.0 | 16.5 | 16.4 | 12.6 | 12.6 | 29.6 | 22.3 | 41.6 | 4.9 | 41.7 | 8.5 | 38.7 | 25.5 | 42.3 | 8.3 | **45.6** | 10.2 | 42.9 | **40.4** | +0.0 | +40.4 |
| | DTD | 42.3 | 0.1 | 29.9 | 29.3 | 35.3 | 34.3 | **44.2** | 36.2 | 40.5 | 26.7 | 40.4 | 8.5 | 40.6 | 22.1 | 41.9 | 20.6 | 35.7 | 22.1 | 41.9 | **39.1** | -0.4 | +39.1 |
| | PCAM | 48.4 | 7.4 | 48.8 | 48.8 | 51.7 | **51.7** | **54.6** | 39.2 | 48.6 | 48.4 | 48.4 | 36.1 | 48.5 | 48.2 | 44.3 | 4.3 | 48.2 | 23.3 | 48.5 | 50.3 | +0.1 | +42.9 |
| All | Avg. | 63.6 | 1.0 | 43.6 | 42.3 | 50.1 | 48.9 | **63.3** | 51.2 | 56.7 | 34.5 | 61.0 | 11.0 | 61.2 | 34.4 | 63.2 | 23.7 | 55.0 | 24.5 | 62.9 | **58.1** | -0.7 | +57.1 |

*Table 2.* Top-1 zero-shot classification accuracy (%) under stronger adversarial attacks (50-step PGD and AutoAttack, $\ell_\infty = 4/255$).

| Method | General | | | Fine-Grained | | | Scene | | | Domain | | |
| | Clean | PGD | AutoAttack | Clean | PGD | AutoAttack | Clean | PGD | AutoAttack | Clean | PGD | AutoAttack |
|---|---|---|---|---|---|---|---|---|---|---|---|---|
| CLIP | 79.4 | 0.0 | 0.0 | 76.2 | 0.0 | 0.0 | 40.3 | 0.0 | 0.0 | 39.2 | 0.0 | 0.0 |
| TeCoA | 66.3 | 65.1 | 65.0 | 39.0 | 35.8 | 35.7 | 21.5 | 20.0 | 19.7 | 25.1 | 25.5 | 25.5 |
| FARE | 66.6 | 65.8 | 65.4 | 59.7 | 55.7 | 55.2 | 26.0 | 24.2 | 24.2 | 28.0 | 27.8 | 27.6 |
| R-TPT | 78.5 | 36.2 | 28.4 | **76.7** | 46.4 | 42.8 | **41.7** | 26.7 | 26.4 | 38.1 | 27.5 | 23.2 |
| TTE | **80.7** | 13.4 | 11.5 | 73.6 | 1.7 | 0.4 | 40.6 | 1.2 | 1.4 | 38.0 | 13.4 | 11.5 |
| LPF | 78.6 | 37.4 | 30.8 | 68.2 | 11.3 | 8.2 | 37.0 | 8.0 | 6.9 | 37.1 | 18.6 | 16.7 |
| HD | 76.9 | 0.5 | 0.1 | 72.8 | 0.0 | 0.0 | 37.4 | 0.0 | 0.0 | 37.3 | 0.1 | 0.0 |
| Anti-Adv | 76.2 | 22.6 | 2.7 | 74.1 | 1.7 | 0.0 | 38.9 | 1.1 | 0.2 | 37.1 | 9.8 | 1.7 |
| TTC | 73.3 | 14.7 | 0.6 | 59.7 | 2.0 | 0.0 | 28.9 | 1.2 | 0.0 | 35.8 | 9.3 | 0.3 |
| CSR | 78.7$^{\downarrow0.7}$ | **78.0**$^{\uparrow78.0}$ | **66.4**$^{\uparrow66.4}$ | 74.8$^{\downarrow1.4}$ | **75.4**$^{\uparrow75.4}$ | **60.3**$^{\uparrow60.3}$ | 39.8$^{\downarrow0.5}$ | **43.8**$^{\uparrow43.8}$ | **34.2**$^{\uparrow34.2}$ | **39.0**$^{\downarrow0.2}$ | **43.2**$^{\uparrow43.2}$ | **32.6**$^{\uparrow32.6}$ |

*Table 3.* Running time per image (ms) (↓) on single RTX 4090.

| Time | CLIP | R-TPT | HD | Anti-Adv | TTE | TTC | CSR(ours) |
|---|---|---|---|---|---|---|---|
| $T_{Clean}$ | 3.37 | 176.45 | 184.43 | 32.39 | 5.43 | 33.24 | **4.16** |
| $T_{Rob.}$ | 3.37 | 176.20 | 189.34 | 32.44 | **5.41** | 35.13 | 26.18 |
| $T_{Avg.}$ | 3.37 | 176.33 | 189.34 | 32.42 | **5.42** | 35.19 | 15.17 |

*Table 4.* Evaluation on different attack objectives, $\ell_\infty = 4/255$.

| Method | Clean | Cross-Modal | | Targeted | | | Label-free | |
| | | PGD | AA | PGD | DLR | AA | PGD | AA |
|---|---|---|---|---|---|---|---|---|
| CLIP | 63.9 | 0.0 | 0.0 | 0.0 | 0.0 | 0.0 | 0.1 | 0.0 |
| FARE | 48.0 | 46.2 | 45.9 | 46.7 | 46.3 | **46.3** | 46.6 | 46.0 |
| TTC | 40.9 | 2.7 | 0.3 | 25.4 | 6.8 | 6.2 | 10.2 | 1.6 |
| CSR | **62.5** | **60.3** | **60.1** | **46.9** | **46.4** | 45.0 | **49.7** | **47.6** |

$\tau = 0.85$, the noise budget is set to $4/255$, with a step size of $2/255$, and the number of iterations to 3. We provide additional experimental details in Appendix F.1.

**Comparison Methods.** We evaluate our proposed method against existing state-of-the-art defenses. The comparison set comprises Test-Time Defense—specifically TTE (Pérez et al., 2021), HD (Wu et al., 2021), Anti-Adv (Alfarra et al., 2022), LPF (Ziyadinov & Tereshonok, 2023), TTC (Xing et al., 2025), and R-TPT (Sheng et al., 2025). Adversarial fine-tuning baselines, such as TeCoA (Mao et al., 2023) and FARE (Schlarmann et al., 2024) (fine-tuned on ImageNet), are also included. Baselines are implemented using their original hyperparameter settings to ensure a fair comparison.

### 5.2 Main Results

**Results on 16 Datasets.** Table 1 presents a comprehensive evaluation of zero-shot classification robustness across 16 datasets. Adversarial fine-tuning (*e.g.*, TeCoA, FARE) suffers from severe benign performance degradation (↓ 20%

as default and also conduct experiments on ViT-B/32, ViT-L/14, and ViT-L/14@336px (LLaVA visual encoder).

**Implementation Details.** Unless otherwise specified, we set the perturbation budget to $1/255$ and the number of attack steps to 10 for PGD (Madry et al., 2017) and 50 for AutoAttack (Croce & Hein, 2020). For our CSR, we set the Gaussian filter radius $r = 40$, the detection threshold

*Table 5.* Comparison of zero-shot classification accuracy on CLIP-B/32 and CLIP-L/14 under 10-step PGD ($\ell_\infty = 1/255$).

| Method | CLIP-B/32 | | | | | | | | CLIP-L/14 | | | | | | | |
|---|---|---|---|---|---|---|---|---|---|---|---|---|---|---|---|---|
| | General | | FG | | Scene | | Domain | | General | | FG | | Scene | | Domain | |
| | clean | rob. | clean | rob. | clean | rob. | clean | rob. | clean | rob. | clean | rob. | clean | rob. | clean | rob. |
| CLIP | 76.7 | 4.0 | 71.8 | 0.3 | 38.8 | 0.4 | 35.9 | 6.6 | 83.7 | 4.0 | 84.2 | 0.3 | 45.3 | 0.2 | 46.8 | 0.3 |
| TeCoA | 60.3 | **58.9** | 35.1 | 32.4 | 17.2 | 17.8 | 25.1 | 25.6 | 73.3 | 72.9 | 45.6 | 41.6 | 26.5 | 25.3 | 30.6 | 31.4 |
| FARE | 59.6 | 58.4 | 52.6 | **48.3** | 23.2 | 22.3 | 27.7 | 28.0 | 75.3 | 74.6 | 63.9 | 57.6 | 32.7 | 32.5 | 32.7 | 32.4 |
| R-TPT | 72.9 | 41.9 | **71.4** | 45.4 | 38.7 | 29.6 | 35.7 | 27.2 | 84.2 | 76.8 | **83.5** | 70.3 | **47.1** | 37.5 | 42.4 | 37.3 |
| LPF | 74.8 | 38.0 | 62.0 | 18.9 | 36.2 | 11.3 | 33.7 | 17.4 | 84.0 | 66.9 | 78.8 | 51.7 | 44.2 | 26.2 | 44.5 | 28.5 |
| HD | 76.0 | 15.9 | 68.6 | 4.9 | 35.1 | 3.1 | 35.1 | 13.6 | 82.7 | 38.3 | 79.7 | 9.3 | 43.0 | 6.2 | 44.2 | 15.9 |
| Anti-Adv | 75.3 | 39.1 | 70.3 | 12.6 | 37.3 | 7.0 | 33.2 | 19.9 | 82.5 | 67.9 | 82.4 | 48.0 | 45.2 | 22.0 | 43.5 | 30.9 |
| TTE | **77.5** | 47.3 | 68.6 | 27.9 | **39.3** | 9.6 | **36.8** | 19.3 | **86.1** | 58.5 | 81.6 | 32.3 | 47.8 | 12.2 | 43.7 | 23.6 |
| TTC | 74.5 | 43.5 | 66.5 | 27.5 | 32.9 | 17.5 | 34.2 | 22.4 | 80.5 | 33.3 | 70.3 | 39.7 | 35.1 | 19.7 | 42 | 15.5 |
| CSR | 76.1$^{\downarrow 0.6}$ | 50.9$^{\uparrow 46.9}$ | **71.4**$^{\downarrow 0.4}$ | 41.1$^{\uparrow 40.8}$ | 38.3$^{\downarrow 0.5}$ | **30.2**$^{\uparrow 29.8}$ | 35.8$^{\downarrow 0.2}$ | **29.7**$^{\uparrow 23.1}$ | 82.2$^{\downarrow 1.5}$ | **77.5**$^{\uparrow 73.5}$ | **83.5**$^{\downarrow 0.7}$ | **73.6**$^{\uparrow 73.3}$ | 44.2$^{\downarrow 1.1}$ | **45.8**$^{\uparrow 45.6}$ | **45.6**$^{\downarrow 1.2}$ | **45.9**$^{\uparrow 45.6}$ |

*(a) General-ImageNet*    *(b) FG-Flower102*    *(c) Scene-SUN397*    *(d) Domain-PCAM*

*Figure 5.* Ablation study on the rectification steps $N$, the Gaussian filter radius $r$, and the detection threshold $\tau$. More in Appendix H

*Table 6.* Ablation study on CSR framework.

| Loss Terms | | Greedy | General | | FG | | Scene | | Domain | |
|---|---|---|---|---|---|---|---|---|---|---|
| Attraction | Repulsion | | Clean | Rob. | Clean | Rob. | Clean | Rob. | Clean | Rob. |
| | ✓ | ✓ | 78.2 | 44.7 | 74.5 | 33.3 | 39.6 | 22.3 | 38.9 | 30.3 |
| ✓ | | ✓ | **79.3** | 51.6 | **75.8** | 33.8 | **40.3** | 20.2 | **39.1** | 23.3 |
| ✓ | ✓ | | 78.6 | 68.5 | 74.7 | 65.1 | 39.6 | 44.1 | 39.0 | 39.9 |
| ✓ | ✓ | ✓ | 78.7 | **69.8** | 74.8 | **65.3** | 39.8 | **44.6** | 39.0 | **40.3** |

*Table 7.* Results on various vision tasks. (mIoU/Accuracy)

| Method | Segmentation | | Captioning | | VQA | |
|---|---|---|---|---|---|---|
| | Clean | Rob. | Clean | Rob. | Clean | Rob. |
| Origin Model | 37.7 | 20.8 | 100.0 | 21.3 | 100.0 | 22.6 |
| +TTC defense | 35.1 | 21.9 | 87.4 | 24.1 | 87.7 | 27.3 |
| +CSR defense | **37.5**$^{\downarrow 0.2}$ | **37.5**$^{\uparrow 16.7}$ | **99.2**$^{\downarrow 0.8}$ | **64.3**$^{\uparrow 43.0}$ | **98.7**$^{\downarrow 1.3}$ | **65.6**$^{\uparrow 43.0}$ |

and ↓ 13%, respectively) due to catastrophic forgetting. Among test-time methods, R-TPT achieves competitive robustness but incurs high inference latency—approximately 12× that of CSR (see Table 3). Notably, passive low-pass filtering (LPF) provides insufficient robustness. TTC, the most relevant baseline, yields only 24.5% accuracy because maximizing divergence without a reliable anchor causes embeddings to drift from the semantic manifold. In contrast, CSR effectively anchors the optimization, achieving **58.1%** robust accuracy—surpassing the strongest baseline

by ↑ **6.9%**—with a negligible 0.7% clean accuracy drop.

**Results under Stronger Attacks.** To rigorously evaluate defense stability, we escalate the threat model to include 50-step PGD and the stronger AutoAttack (AA) with an increased budget of $\ell_\infty = 4/255$. As detailed in Table 2, CSR demonstrates exceptional resilience, consistently outperforming baselines while preserving clean accuracy. Adversarial fine-tuning methods (*e.g.*, TeCoA, FARE) exhibit commendable robustness under strong attacks, validating the efficacy of weight modification; however, this comes at the cost of severe degradation in benign performance. In contrast, other test-time defenses crumble under increased adversarial pressure. Methods like TTC, Anti-Adv, and HD exhibit near-zero robustness against AA, revealing their fragility. Even the computationally expensive R-TPT struggles to maintain stability (dropping to 28.4% on General). Conversely, CSR effectively rectifies the adversarial manifold, achieving **66.4%** average accuracy under AA—a significant **18.1%** improvement over the best test-time baseline—while preserving original zero-shot capabilities.

**Results under Different Attack Objectives.** To validate the versatility of CSR, we extend our evaluation to a broad spectrum of adversarial objectives (Details in Appendix F.2),

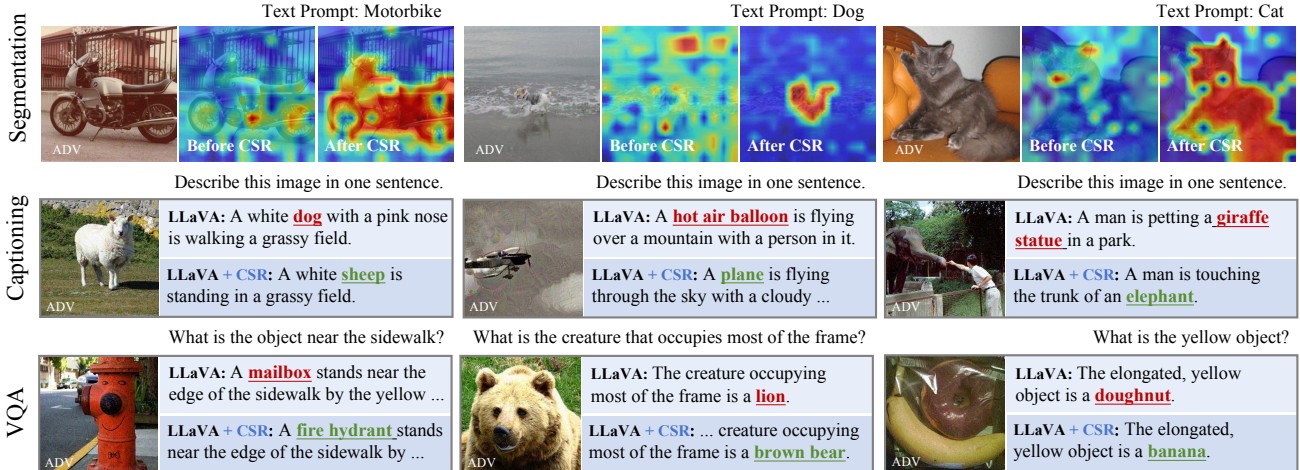

*Figure 6.* Examples of Segmentation Heatmaps (top), Image Captioning, and Visual Question Answering (bottom). More in Appendix J.

including Cross-Modal attacks (disrupting image-text alignment), Targeted attacks (forcing specific misclassification), and Label-free attacks (maximizing visual feature deviation). As presented in Table 4, CSR exhibits consistent superiority across all scenarios. Notably, in Cross-Modal settings, CSR significantly outperforms FARE by approximately ↑ **14%** under PGD attacks, while maintaining high benign accuracy. These confirm that CSR functions as a generalized defense, capable of effectively rectifying adversarial perturbations agnostic to the attacker's specific optimization strategy.

**Inference Efficiency Analysis.** Table 3 compares the inference latency across different defense methods. While R-TPT achieves competitive robustness, it incurs high computational costs (176.33 ms) due to complex image transformations and prompt optimization. In contrast, CSR demonstrates superior efficiency. This confirms that CSR achieves the optimal trade-off between performance and computational cost, making it suitable for real-time deployment.

**Results on Different Backbones.** To validate the architectural universality of CSR, we extend our evaluation to CLIP-ViT-B/32 and CLIP-ViT-L/14. As shown in Table 5, the observations remain consistent with those on ViT-B/16. CSR demonstrates superior scalability, achieves state-of-the-art robustness while preserving high clean accuracy.

**Results on Various Vision Tasks.** Most existing test-time defenses are inherently task-specific, rendering them inapplicable to complex downstream scenarios. To demonstrate the universality of CSR, we subject the CLIP-based Zero-Shot Segmentation (Lan et al., 2024) to PGD attacks ($\ell_\infty = 4/255$) and evaluate the robustness of LLaVA on Captioning and VQA tasks against strong V-Attack (Nie et al., 2025) ($\ell_\infty = 16/255$). Detailed experiment settings are provided in Appendix F.3. As reported in Table 7, while the generic method TTC fails to provide an effective defense, yielding negligible improvements, CSR serves as a robust shield for various downstream applications. It remarkably restores segmentation

performance (boosting mIoU by ↑ **16.7%**) and successfully mitigates adversarial attacks in LLaVA, improving accuracy by ↑ **43%** on both Captioning and VQA tasks. We supplement the results of Qwen2.5-VL in Appendix F.3. These quantitative gains are corroborated qualitatively in Figure 6. In the top row, CSR rectifies corrupted segmentation masks to recover fine-grained structural details; in the bottom row, it effectively purifies the visual input, correcting model errors (*e.g.*, rectifying "dog" back to "sheep") and ensuring accurate reasoning. This confirms that CSR functions as a versatile, plug-and-play module that promotes broad security across the CLIP ecosystem (*e.g.*, Large Vision-Language Models).

### 5.3 Ablation

**The CSR Framework.** We validate the contribution of each component within our proposed framework, as summarized in Table 6. The results indicate that the Attraction and Repulsion terms are mutually reinforcing. Employing either term in isolation yields suboptimal robustness. However, their joint application leads to a substantial performance leap (*e.g.*, boosting robustness on General datasets from **51.6%** to **69.8%**). Furthermore, the incorporation of the Greedy Selection strategy consistently optimizes the results.

**Hyperparameter Sensitivity.** Figure 5 illustrates CSR's sensitivity to the rectification steps $N$, the Gaussian filter radius $r$, and the detection threshold $\tau$. Specifically, increasing $N$ consistently enhances robustness but scales up computational overhead; we find $N = 3$ strikes an optimal balance between efficacy and efficiency. Remarkably, under strong attacks ($\ell_\infty = 4/255$), the post-rectification accuracy even surpasses the clean accuracy. This counterintuitive phenomenon suggests that adversarial examples implicitly encode directional priors of the decision boundary, which CSR leverages to bolster robustness. Notably, a small radius (*e.g.*, $r = 10$) yields suboptimal robustness results. We attribute this to excessive information loss in the anchor, which weakens semantic guidance and hinders adversar-

*Table 8.* Top-1 accuracy (%) under two boundary-case attacks that lie outside the standard $\ell_\infty$ additive setting.

| Method | Clean | Patch | Color |
|--------|-------|-------|-------|
| CLIP | 63.9 | 0.0 | 2.2 |
| TTC | 40.9 | 30.0 | 6.4 |
| **CSR** | **62.5** | **57.3** | 3.7 |

ial rectification. However, benign samples remain largely unaffected due to our input-adaptive gating mechanism.

# 6 Limitation

CSR assumes that adversarial perturbations predominantly reside in the mid-to-high frequency band, and its effectiveness therefore hinges on the validity of this spectral assumption. To probe its boundary, we evaluate two atypical attacks beyond the standard $\ell_\infty$-bounded additive setting: a targeted localized patch attack (Karmon et al., 2018) crafting a $32 \times 32$ adversarial corner patch (100 steps, lr 0.05), and a global color attack (Zhao et al., 2023b) that optimizes per-channel piecewise color curves in an explicit color filter space ($K$=64, 50 steps). As reported in Table 8, CSR remains highly effective under localized patch attacks (**57.3%** vs. 30.0% for TTC), since their spatially-confined signal is largely suppressed by low-pass filtering, leaving the global low-frequency anchor reliable. In contrast, global color attacks pose a genuinely harder boundary case (CSR: 3.7%): such transformations preserve their deceptive cues even after low-pass filtering, contaminating the semantic anchor itself. We view defending against this class of color-space manipulations—where the spectral assumption no longer holds—as an important direction for future work.

# 7 Conclusion

In this work, we deconstruct the vulnerability of CLIP through the lens of frequency analysis. Guided by these insights, we introduce Contrastive Spectral Rectification (CSR), a novel test-time defense that strategically leverages contrastive anchors for both adversarial detection and purification. Given that CLIP serves as the foundational "eyes" of modern multimodal systems, ensuring its security is paramount. As an efficient, effective, and universal framework, CSR not only fortifies the visual encoder itself but also provides a scalable blueprint for enhancing the robustness of the broader ecosystem of vision-language tasks, paving the way for more reliable and secure artificial intelligence.

# Impact Statement

This work advances the field of adversarial robustness by addressing the critical security vulnerabilities of foundational Vision-Language Models (VLMs). As VLMs increasingly serve as the perceptual backbone for diverse intelligent systems, their susceptibility to adversarial attacks poses a systemic risk to the broader AI ecosystem. The distinct impact of our proposed Contrastive Spectral Rectification

(CSR) lies in its universality; unlike task-specific defenses, CSR provides a scalable, task-agnostic protection mechanism. We demonstrate its broad efficacy across complex downstream tasks, including Semantic Segmentation, Image Captioning, and Visual Question Answering (VQA). By introducing a training-free, test-time defense, this work facilitates the deployment of secure AI in safety-critical applications, ensuring that the generalization power of VLMs is maintained without compromising reliability.

# Acknowledgements

This work is partially supported by the Strategic Priority Research Program of the Chinese Academy of Sciences under Grant XDB0680202, the Key Research and Development Program of Xinjiang Uyghur Autonomous Region under Grant 2024B03026, and Beijing Nova Program under Grant 20230484368.

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

## Appendix: Table of Contents

## A    Related Works

**Adversarial Attacks on VLMs.** Current research on VLM vulnerabilities can be broadly categorized into white-box optimization and black-box transfer-based paradigms (Qian et al., 2025; Chen et al., 2025b; Liu et al., 2025; Zhang et al., 2025b). In the white-box setting, Typographical Attacks (Noever & Noever, 2021) exposed CLIP's "read first, look later" bias via visible text patches. Optimization-based strategies subsequently emerged to disrupt cross-modal alignment: Co-Attack (Zhang et al., 2022) pioneered the joint perturbation of both visual and textual inputs, while AdvCLIP (Zhou et al., 2023) explored universal perturbations capable of deceiving diverse downstream tasks. To address black-box transferability, recent works predominantly leverage augmentation and cross-modal priors. Specifically, SGA (Lu et al., 2023), SA-Attack (He et al., 2023), and DI-RAT (Gao et al., 2024) enhance transferability via set-level augmentations. VLP-Attack (Wang et al., 2025b) utilizes contrastive objectives to generate transferable examples, while TMM (Wang et al., 2024a) and VLATTACK (Yin et al., 2023) exploit modality-consistent features and combined multi-modal perturbations. Additionally, PRM (Hu et al., 2024) specifically targets vulnerabilities in downstream applications. C-PGC (Fang et al., 2024) pioneers this line of research by optimizing Universal Adversarial Perturbations (UAPs) via contrastive learning, and ETU (Zhang et al., 2024d) further elevates their potency through global optimization and mix-based data augmentation.

**Adversarial Attacks on LVLMs.** A critical vulnerability in Large Vision-Language Models (LVLMs)—such as LLaVA (Liu et al., 2023), Qwen-VL (Bai et al., 2023), InternVL (Chen et al., 2024), and DeepseekVL (Lu et al., 2024)—stems from their vision encoders. This architectural commonality allows attackers to utilize CLIP as a surrogate model to craft transfer-based attacks (Ma et al., 2025). For instance, AttackBard (Dong et al., 2023), AttackVLM (Zhao

et al., 2023a), M-Attack (Li et al., 2025), FOA-Attack (Jia et al., 2025), and V-Attack (Nie et al., 2025) demonstrate that adversarial examples generated from CLIP can effectively mislead closed-source models. Generator-based strategies leverage generative priors: AdvDiffVLM (Guo et al., 2024) employs diffusion models with gradient estimation to synthesize adversarial images, while AnyAttack (Zhang et al., 2024c) introduces a self-supervised contrastive framework to generate label-free adversarial examples capable of compromising diverse tasks.

Given that CLIP-based encoders act as the cornerstone for modern multimodal systems, their adversarial fragility constitutes a critical security bottleneck. Therefore, developing robust defenses for these foundational models is pivotal to securing the broader landscape of intelligent systems against adversarial threats. This motivates our CSR defense.

**Frequency and Adversarial Robustness.** A growing line of work investigates the interplay between frequency components and adversarial robustness, generally observing that low-frequency features are more robust than high-frequency ones. Garg et al. (Garg et al., 2018) characterize adversarially robust features from a spectral viewpoint and propose a training-time learning scheme to extract them; Wang et al. (Wang et al., 2020) attribute the generalization gap of CNNs to their reliance on high-frequency components and analyze its consequences for robustness; Bu et al. (Bu et al., 2023) build more robust models by explicitly biasing the network toward low-frequency information during training. CSR departs from this body of work in two key respects. First, in terms of methodology, both (Garg et al., 2018) and (Bu et al., 2023) improve robustness through training-time learning or model-side design, whereas CSR is a training-free, plug-and-play test-time defense that requires neither retraining nor architectural modification. Second, in terms of scope and contribution, (Wang et al., 2020) confines its analysis to CNNs, while CSR revisits the frequency–robustness relationship in pretrained VLMs and further connects frequency properties to feature stability through both empirical and theoretical analyses (Section 3). Building on these insights, CSR realizes a practical defense via input-adaptive contrastive rectification rather than analysis alone.

## B    Spectral Vulnerability Across Backbones

The main text observes a consistent mid-to-high frequency vulnerability across CLIP-B/32, CLIP-B/16, CLIP-L/14, and the CLIP-L/14@336 encoder used in LLaVA, and Qwen2.5-VL's self-trained visual encoder follows the same trend (Appendix F.3). To quantify this, we measure the gradient energy distribution across frequency bands on ImageNet classification. As reported in Table 9, roughly 90% of the adversarial gradient energy concentrates in the mid-to-high frequency bands for every tested backbone, confirming

*Table 9.* Gradient distribution ratio (%) across frequency bands on ImageNet. Across all visual backbones, the adversarial gradient energy is dominated by the mid-to-high frequency bands.

| | CLIP-B/32 | CLIP-B/16 | CLIP-L/14 | CLIP-L/14@336 |
|---|---|---|---|---|
| Low frequency | 9.79 | 11.02 | 10.74 | 5.23 |
| Mid-to-high frequency | **90.21** | **88.98** | **89.26** | **94.77** |

that the spectral vulnerability CSR exploits is a shared property of large-scale pre-trained visual encoders.

## C  Theoretical Analysis of Gradient Conflict

In this section, we theoretically analyze why limiting adversarial attacks to the low-frequency band is inherently inefficient. This inefficiency arises from a fundamental geometric conflict: the gradient direction required to maximize adversarial loss exhibits a consistent negative cosine similarity with the gradient of low-frequency constraints. In the following derivation, we prove this by analyzing the geometric relationship between the proposed spectral consistency objective (which enforces low-frequency alignment) and the direction of adversarial perturbations. By employing a local linearization of the encoder, we show that the optimization landscape inherently suppresses perturbations in the low-frequency band, thereby forcing effective attacks to gravitate toward high-frequency components.

**Problem Setup.** Let $f : \mathbb{R}^d \to \mathbb{R}^k$ be the encoder, and $\mathbf{G} \in \mathbb{R}^{d \times d}$ be a symmetric, linear low-pass filter (*e.g.*, Gaussian). We define the high-pass projection operator as $\mathbf{P} = \mathbf{I} - \mathbf{G}$. The consistency loss is given by:

$$\mathcal{L}_{sim}(\boldsymbol{x}) = \frac{1}{2} \|f(\boldsymbol{x}) - f(\mathbf{G}\boldsymbol{x})\|_2^2. \tag{8}$$

**Local Linearization.** Since natural images are dominated by low-frequency components, the term $\|\boldsymbol{x} - \mathbf{G}\boldsymbol{x}\|_2$ is typically small. We approximate the behavior of $f$ using a first-order Taylor expansion around the smoothed anchor point $\mathbf{G}\boldsymbol{x}$. Assuming the encoder is locally smooth:

$$f(\boldsymbol{x}) \approx f(\mathbf{G}\boldsymbol{x}) + \mathbf{J}(\mathbf{G}\boldsymbol{x})(\boldsymbol{x} - \mathbf{G}\boldsymbol{x}) = f(\mathbf{G}\boldsymbol{x}) + \mathbf{J}\mathbf{P}\boldsymbol{x}, \tag{9}$$

where $\mathbf{J} \triangleq \mathbf{J}_f(\mathbf{G}\boldsymbol{x}) \in \mathbb{R}^{k \times d}$ is the input-output Jacobian evaluated at the smoothed image. Under this approximation, the loss simplifies to a quadratic form in the projected space:

$$\tilde{\mathcal{L}}_{sim}(\boldsymbol{x}) = \frac{1}{2} \|\mathbf{J}\mathbf{P}\boldsymbol{x}\|_2^2. \tag{10}$$

**Gradient Derivation.** We derive the gradient of $\tilde{\mathcal{L}}_{sim}$ with respect to $\boldsymbol{x}$. Consistent with the local linearization assumption, we treat the Jacobian $\mathbf{J}$ as a constant linear operator within the neighborhood of $\boldsymbol{x}$. Using the symmetry of $\mathbf{P}$:

$$\nabla_{\boldsymbol{x}}\tilde{\mathcal{L}}_{sim} \approx \mathbf{P}^\top \mathbf{J}^\top \mathbf{J}\mathbf{P}\boldsymbol{x} = \mathbf{P}\mathbf{M}\mathbf{P}\boldsymbol{x}, \tag{11}$$

where $\mathbf{M} = \mathbf{J}^\top \mathbf{J} \in \mathbb{R}^{d \times d}$ is the local metric tensor (or the input-output Jacobian Gram matrix), which characterizes the sensitivity of the feature space to input variations.

**Gradient Conflict and Attack Inefficiency.** Consider an adversarial example $\boldsymbol{x}_{adv} = \boldsymbol{x}_{clean} + \boldsymbol{\delta}$. In optimization-based attacks (*e.g.*, PGD), the perturbation $\boldsymbol{\delta}$ aligns with the adversarial gradient $\nabla_{\boldsymbol{x}}\mathcal{L}_{adv}$ to maximize model error. As established in Section 3.2, the model's Jacobian spectrum is biased toward high frequencies, implying that the most aggressive attack components naturally reside in the high-frequency subspace (*i.e.*, $\mathbf{P}\boldsymbol{\delta} \approx \boldsymbol{\delta}$).

We now analyze the geometric alignment between the gradient of the low-frequency constraint, denoted as $\mathbf{g}_{low} = -\nabla_{\boldsymbol{x}}\tilde{\mathcal{L}}_{sim}$ (which acts as a restorative force towards the natural manifold), and the optimal attack direction $\boldsymbol{\delta}$. Deriving the inner product:

$$\begin{aligned} \langle \mathbf{g}_{low}, \boldsymbol{\delta} \rangle &= \langle -\nabla_{\boldsymbol{x}}\tilde{\mathcal{L}}_{sim}, \boldsymbol{\delta} \rangle \\ &\approx -\boldsymbol{\delta}^\top (\mathbf{P}\mathbf{M}\mathbf{P}) \boldsymbol{\delta} \\ &= -\|\mathbf{J}(\mathbf{P}\boldsymbol{\delta})\|_2^2 < 0. \end{aligned} \tag{12}$$

This strict inequality implies a **consistent negative cosine similarity** between the low-frequency constraint and the adversarial objective. Mathematically, the constraint gradient opposes the direction required for an effective attack.

## D  Spectral Consistency Disparity on 15 Datasets

In this section, we provide a comprehensive evaluation of feature consistency under progressive frequency attenuation across a broader range of datasets and attack configurations to reinforce the observations in Section 3.1. We extend our analysis to include:

- General Classification: CIFAR-10/100, STL-10, Caltech-101/256, and ImageNet.
- Fine-Grained Recognition: Oxford Pets, Flowers-102, Food-101, and Stanford Cars.
- Scene Understanding: SUN397 and Country211.
- Specialized Domains: FGVC-Aircraft, EuroSAT, DTD, and PCAM.

For each dataset, we randomly sample 300 instances and generate adversarial examples (AEs) using 10-step PGD and 30-step AutoAttack under various $\ell_\infty$ budgets. As shown in Figure 9, benign and adversarial samples exhibit a clear distinction across all datasets.

## E  CSR Detection AUC on 15 Datasets

We report the Area Under the Curve (AUC) results for benign and adversarial samples (10-step PGD, $\ell_\infty = 1/255$)

across 15 datasets, using the default hyperparameters: Gaussian filter radius $r = 40$ and detection threshold $\tau = 0.85$. As illustrated in Figure 10, the Area Under the ROC curve exceeds **0.95** across all datasets, with a dense concentration between **0.98** and **0.99**. These results demonstrate that the input-adaptive gating mechanism in CSR—derived from the intrinsic properties of CLIP discussed in Section 3—effectively distinguishes between benign and adversarial inputs. Notably, this mechanism incurs minimal computational overhead (Table 3), introducing negligible latency on benign samples (increasing from 3.37ms to 4.16ms), which underscores its practical utility for real-world deployment.

## F  Additional Details of the Experiment

### F.1  Additional Implementation Details

In our experiments, unless otherwise specified, we employ PGD and AutoAttack (specifically the APGD algorithm) under the $\ell_\infty$ norm constraint. The former serves as a standard benchmark for general evaluation, while the latter provides a more rigorous assessment of robustness under strong attacks. All experiments are conducted on a server equipped with eight NVIDIA GeForce RTX 4090 GPUs.

### F.2  Attacks with Different Objectives

To further validate the versatility and robustness of CSR, we extend our evaluation to a broad spectrum of adversarial objectives beyond standard untargeted attacks. For all experiments in this section, we maintain a consistent adversarial budget of $\epsilon = 4/255$ under the $\ell_\infty$ norm, with the number of attack steps set to $T = 50$. The specific formulations for each objective are detailed as follows:

**Cross-Modal Attacks.**  The objective of cross-modal attacks is to disrupt the inherent semantic alignment between the visual and textual modalities in CLIP's latent space. We employ PGD and AutoAttack (AA) to minimize the cosine similarity between the perturbed image embedding and its corresponding ground-truth text embedding. The loss function is formulated as:

$$\mathcal{L}_{cm} = \frac{f(\boldsymbol{x} + \boldsymbol{\delta})^\top g(\boldsymbol{t}_y)}{\|f(\boldsymbol{x} + \boldsymbol{\delta})\|_2 \cdot \|g(\boldsymbol{t}_y)\|_2} = \boldsymbol{z}_v'^\top \boldsymbol{z}_{t,y}, \quad (13)$$

where $\boldsymbol{z}_v'$ denotes the normalized embedding of the adversarial image. By minimizing this loss, the attacker forces the visual feature to drift away from its semantic anchor.

**Targeted Attacks.**  In the targeted setting, the attacker aims to steer the model's prediction toward a specific, incorrect class $y_{target} \neq y$. This objective is more aggressive than untargeted attacks as it requires the perturbation to encode structured, misleading semantic information. We evaluate CSR against three representative targeted strategies:

- **PGD:** The perturbation is optimized by minimizing the cross-entropy loss between the predicted probability distribution and the one-hot encoded target label:

$$\begin{aligned}\mathcal{L}_{tar} &= -\log p(y_{target} \mid \boldsymbol{x} + \boldsymbol{\delta}) \\ &= -\log\left(\frac{\exp(\tau \cdot \boldsymbol{z}_v'^\top \boldsymbol{z}_{t,y_{target}})}{\sum_{j=1}^{K}\exp(\tau \cdot \boldsymbol{z}_v'^\top \boldsymbol{z}_{t,j})}\right),\end{aligned} \quad (14)$$

where $\boldsymbol{z}_v' = f(\boldsymbol{x} + \boldsymbol{\delta})/\|f(\boldsymbol{x} + \boldsymbol{\delta})\|_2$. This forces the visual feature to align closely with the target class's text embedding $\boldsymbol{z}_{t,y_{target}}$.

- **DLR:** Moreover, we utilize the targeted version of the Difference of Logits Ratio (DLR) loss:

$$\mathcal{L}_{tar\_dlr} = -\frac{s_y - s_{y_{target}}}{s_{\pi_1} - \frac{1}{2}(s_{\pi_3} + s_{\pi_4})}. \quad (15)$$

Here, $s_i = \boldsymbol{z}_v'^\top \boldsymbol{z}_{t,i}$ represents the cosine similarities (logits), and $\pi$ denotes the descending order of these similarities. This loss specifically penalizes the margin between the ground-truth class and the target class.

- **AutoAttack:** We also employ the targeted version of AutoAttack, which serves as a parameter-free benchmark consisting of multiple iterations of targeted APGD to ensure a reliable evaluation of CSR's defense ceiling.

**Label-free Attacks.**  This category represents a label-free setting where the attacker lacks access to ground-truth labels or textual prompts. In this scenario, the attacker aims to destroy the model's representational integrity by maximizing the semantic deviation from the original image. Specifically, we employ PGD and AutoAttack (AA) to minimize the cosine similarity between the perturbed visual embedding and the original, unperturbed embedding:

$$\mathcal{L}_{labfree} = \frac{f(\boldsymbol{x} + \boldsymbol{\delta})^\top f(\boldsymbol{x})}{\|f(\boldsymbol{x} + \boldsymbol{\delta})\|_2 \cdot \|f(\boldsymbol{x})\|_2} = \boldsymbol{z}_v'^\top \boldsymbol{z}_v, \quad (16)$$

where $\boldsymbol{z}_v$ and $\boldsymbol{z}_v'$ denote the normalized visual embeddings of the clean and adversarial images, respectively. By minimizing this objective, the attacker pushes the adversarial feature as far as possible from its original position in the latent space. Since these attacks are agnostic to textual guidance, they rely solely on the intrinsic visual features, making them a robust measure of CSR's ability to rectify samples.

As presented in Table 4, our CSR exhibits consistent superiority across all evaluated scenarios.

### F.3  CSR on Various Vision Tasks

Current test-time defense strategies are often restricted to specific tasks, such as classification, which limits their applicability and hinders the advancement of security across broader vision tasks. In contrast, our proposed CSR is task-agnostic and can be widely applied to bolster general visual

security. To demonstrate this versatility, we extend our evaluation to three additional tasks: Semantic Segmentation, Image Captioning, and Visual Question Answering (VQA).

For the semantic segmentation task, we conduct experiments using the CLIP-B/16 backbone. Regarding the dataset, we utilize VOC2010, from which we randomly select 3k images to perform evaluation. Adversarial examples are generated via AutoAttack (specifically the APGD algorithm) under an $\ell_\infty$ constraint of $4/255$ for $T = 50$ iterations. Following the unlabeled attack setting, the optimization objective is to push the adversarial features away from the original image embeddings in the latent space. For the segmentation framework, we adopt the training-free strategy proposed in (Lan et al., 2024), which enables zero-shot semantic segmentation leveraging CLIP's cross-modal capabilities.

For Image Captioning and Visual Question Answering (VQA) tasks, we conduct evaluations using the LLaVA framework. To rigorously assess the rectification efficacy of CSR, adversarial examples are generated via V-Attack (Nie et al., 2025), a state-of-the-art attack strategy for Large Vision-Language Models (LVLMs). Specifically, V-Attack utilizes CLIP-L/14@336px as a surrogate model to produce adversarial perturbations that exhibit high transferability across diverse tasks, models, and prompts. Following the configuration in V-Attack, we set the noise budget to $\ell_\infty = 16/255$ with 200 iterations to ensure a sufficiently strong attack. The evaluation is performed on a subset of 300 samples randomly sampled from the COCO dataset. Since V-Attack targets a specific object within an image per iteration, we employ a Large Language Model (LLM) as the evaluator to determine attack success, strictly adhering to the experimental protocols established in V-Attack.

To further verify that the applicability of CSR extends to a wider family of Large Vision-Language Models (LVLMs), we conduct an additional evaluation on Qwen2.5-VL under the standard M-Attack (Li et al., 2025) protocol. M-Attack crafts highly transferable adversarial examples through an ensemble of surrogate models and has been shown to compromise even strong black-box models. We follow its default configuration with a perturbation budget of $\ell_\infty = 16/255$ and adopt the same captioning evaluation protocol used for the LLaVA experiments above. As reported in Table 10, while the generic test-time defense TTC yields only a marginal gain ($18.0\% \rightarrow 21.0\%$), CSR substantially restores the captioning accuracy of Qwen2.5-VL from $18.0\%$ to $\mathbf{57.0}\%$, corroborating that CSR generalizes across both VLM backbones and attack protocols.

As reported in Table 7, CSR demonstrates robust performance across all three aforementioned tasks, with nearly negligible degradation in accuracy on benign samples. These results indicate that CSR can extensively bolster the security of various vision tasks, providing a solid foundation

*Table 10.* Captioning accuracy (%) on Qwen2.5-VL under the standard M-Attack (Li et al., 2025) protocol ($\ell_\infty = 16/255$).

| Method | Accuracy ($\uparrow$) |
|---|---|
| Qwen2.5-VL | 18.0 |
| Qwen2.5-VL + TTC | 21.0 |
| Qwen2.5-VL + CSR | **57.0** |

*Table 11.* Classification accuracy (%) of CSR under different detection thresholds $\tau$, with the Gaussian filter radius fixed at $r = 40$. CSR remains stable across the swept interval.

| $\tau$ | 0.78 | 0.80 | 0.82 | 0.84 | 0.86 |
|---|---|---|---|---|---|
| Clean | 63.7 | 63.7 | 63.7 | 63.1 | 62.5 |
| APGD ($2/255$) | 64.1 | 64.6 | 65.0 | 65.1 | 65.2 |

and significant implications for future research in adversarial attacks and defenses.

# G  Results on CLIP ViT-L/14 and ViT-B/32

In this section, we present the fine-grained results across 16 datasets on CLIP-ViT-L/14 and CLIP-ViT-B/32 backbones in Tables 12 and 13, respectively. As a detailed extension of the main results in Table 5, these evaluations corroborate the architectural universality and scalability of the proposed CSR method, aligning with the findings for ViT-B/16 previously discussed in the main text.

# H  Ablation of Rectification Steps

As illustrated in Figure 8, we present additional ablation studies on the number of rectification steps $N$ across other datasets. Consistent with the analysis in the main text, increasing $N$ progressively enhances the robustness of CSR, albeit at the cost of higher inference latency.

# I  Ablation of Detection Threshold $\tau$

Since the gating decision in CSR directly governs whether the rectification process is triggered, we examine the sensitivity of the overall framework to the detection threshold $\tau$. With the Gaussian filter radius fixed at $r = 40$, we sweep $\tau$ from 0.78 to 0.86 and evaluate both clean accuracy and adversarial robustness under APGD ($\ell_\infty = 2/255$). As reported in Table 11, CSR remains stable across the entire interval: clean accuracy fluctuates within $1.2\%$ ($63.7\% \rightarrow 62.5\%$) while robust accuracy varies by only $1.1\%$ ($64.1\% \rightarrow 65.2\%$). This stability stems from the pronounced spectral consistency gap between benign and adversarial samples, which leaves a wide operational margin for $\tau$ and obviates the need for per-domain tuning.

# J  Additional Examples

To complement Table 4 and Figure 6, we provide additional examples in Figure 7.

*Table 12.* Performance comparison across different dataset types and method categories on **CLIP-L/14**.

| | Dataset | Original | | Adversarial Fine-tuning | | | | Test-Time Defense | | | | | | | | | | | | | | | Δ | |
| | | CLIP | | TeCoA | | FARE | | R-TPT | | LPF | | HD | | Anti-Adv | | TTE | | TTC | | CSR (Ours) | | | |
| Type | Name | Clean | Rob. | Clean | Rob. | Clean | Rob. | Clean | Rob. | Clean | Rob. | Clean | Rob. | Clean | Rob. | Clean | Rob. | Clean | Rob. | Clean | Rob. | Clean | Rob. |
|---|---|---|---|---|---|---|---|---|---|---|---|---|---|---|---|---|---|---|---|---|---|---|---|
| General | ImageNet | 69.9 | 0.6 | 69.2 | 67.5 | 63.9 | 61.5 | **71.2** | 60.0 | 64.2 | 43.2 | 66.4 | 10.1 | 68.0 | 37.1 | 71.5 | 32.4 | 54.7 | 35.8 | 68.0 | **64.1** | -1.9 | +63.5 |
| | CIFAR10 | 93.4 | 1.2 | 73.7 | 73.7 | 76.6 | 76.5 | 90.2 | 82.1 | 94.3 | 71.0 | 92.1 | 40.5 | 89.3 | 78.5 | 92.3 | 47.0 | **94.1** | 18.4 | 90.7 | **83.6** | -2.7 | +82.4 |
| | CIFAR100 | 65.0 | 0.1 | 44.0 | 43.8 | 45.9 | 45.7 | 65.7 | **58.3** | 72.8 | 40.3 | 67.9 | 26.7 | 64.7 | 51.8 | **72.9** | 37.9 | 71.0 | 7.0 | 62.9 | 60.5 | -2.1 | +60.4 |
| | STL10 | 99.4 | 14.1 | 93.1 | 93.1 | 96.2 | 96.1 | 98.8 | **93.4** | 99.2 | 91.7 | 98.6 | 67.8 | 99.0 | 93.3 | 98.6 | 88.9 | 99.4 | 53.2 | **99.2** | 91.2 | -0.2 | +77.1 |
| | Caltech101 | 86.2 | 4.5 | 80.8 | 80.8 | 85.3 | 85.0 | 89.0 | 81.6 | 86.3 | 77.5 | 84.7 | 45.2 | 86.0 | 72.6 | **91.2** | 73.5 | 81.6 | 41.6 | 85.3 | **82.1** | -0.9 | +77.6 |
| | Caltech256 | 88.3 | 3.7 | 79.2 | 78.7 | 83.9 | 82.8 | **90.5** | **85.4** | 87.4 | 77.9 | 86.2 | 39.5 | 88.0 | 74.3 | 89.9 | 71.3 | 82.0 | 44.0 | 87.3 | 83.2 | -1.0 | +79.5 |
| Fine-G | OxfordPets | 93.5 | 0.3 | 76.1 | 72.9 | 86.6 | 79.4 | **94.2** | **81.4** | 88.3 | 61.7 | 90.5 | 13.9 | 91.7 | 61.1 | 90.1 | 36.0 | 84.8 | 46.8 | 93.1 | 76.4 | -0.4 | +76.1 |
| | Flowers102 | 73.5 | 0.5 | 35.6 | 34.0 | 55.0 | 50.8 | 71.5 | 60.8 | 72.5 | 50.1 | 70.9 | 10.3 | **73.5** | 46.3 | 73.0 | 34.9 | 66.0 | 33.6 | 73.3 | **67.0** | -0.2 | +66.5 |
| | Food101 | 90.8 | 0.1 | 36.2 | 34.8 | 49.4 | 48.0 | **90.6** | 79.3 | 86.8 | 61.2 | 88.7 | 7.0 | 87.7 | 53.1 | 89.8 | 36.8 | 68.3 | 47.9 | **90.6** | **85.8** | -0.2 | +85.7 |
| | StanfordCars | 79.1 | 0.2 | 34.5 | 24.7 | 64.7 | 52.3 | **77.5** | 59.8 | 67.6 | 33.9 | 68.8 | 5.8 | 76.8 | 31.6 | 73.4 | 21.4 | 61.9 | 30.3 | 76.9 | **65.1** | -2.2 | +64.9 |
| Scene | SUN397 | 66.8 | 0.3 | 47.8 | 45.8 | 57.1 | 57.7 | **69.9** | 61.5 | 65.6 | 45.3 | 64.9 | 11.5 | 68.5 | 38.0 | 69.4 | 22.1 | 56.4 | 33.0 | 65.3 | **65.2** | -1.5 | +64.9 |
| | Country211 | 23.7 | 0.0 | 5.2 | 4.7 | 8.3 | 7.3 | 24.2 | 13.4 | 22.8 | 7.1 | 21.1 | 0.8 | 21.9 | 6.0 | **26.2** | 2.3 | 13.8 | 6.3 | 23.0 | **26.4** | -0.7 | +26.4 |
| Domain | FGVCAircraft | 29.4 | 0.0 | 10.6 | 15.1 | 21.3 | 22.2 | **33.2** | 20.7 | 26.1 | 11.1 | 26.5 | 1.4 | 26.5 | 10.7 | 30.2 | 5.3 | 23.4 | 12.1 | 25.8 | **32.5** | -3.6 | +32.5 |
| | EuroSAT | 54.8 | 0.1 | 19.3 | 19.2 | 12.3 | 12.3 | 38.8 | 33.6 | 53.2 | 19.0 | 47.5 | 11.5 | 46.9 | 30.4 | 41.1 | 12.0 | **52.1** | 7.7 | 54.7 | **41.3** | -0.1 | +41.2 |
| | DTD | 53.2 | 0.7 | 33.2 | 32.5 | 37.3 | 35.8 | **53.8** | 44.4 | 48.9 | 35.2 | 52.3 | 15.1 | 50.3 | 33.5 | 52.5 | 26.6 | 42.4 | 25.6 | 50.9 | **50.4** | -2.3 | +49.7 |
| | PCAM | 49.6 | 0.2 | 59.1 | 58.7 | **59.7** | 59.4 | 43.6 | 50.4 | 49.9 | 48.8 | 50.4 | 35.4 | 50.2 | 49.1 | 50.8 | 50.6 | 50.1 | 16.6 | 51.0 | 59.2 | +1.4 | +59.0 |
| All | Avg. | 69.8 | 1.7 | 49.9 | 48.8 | 56.5 | 54.6 | **68.9** | 60.4 | 67.9 | 48.0 | 67.3 | 21.4 | 68.1 | 54.2 | 69.6 | 37.4 | 62.6 | 28.7 | 68.6 | **64.6** | -1.2 | +62.9 |

*Table 13.* Performance comparison across different dataset types and method categories on **CLIP-B/32**.

| | Dataset | Original | | Adversarial Fine-tuning | | | | Test-Time Defense | | | | | | | | | | | | | | | Δ | |
| | | CLIP | | TeCoA | | FARE | | R-TPT | | LPF | | HD | | Anti-Adv | | TTE | | TTC | | CSR (Ours) | | | |
| Type | Name | Clean | Rob. | Clean | Rob. | Clean | Rob. | Clean | Rob. | Clean | Rob. | Clean | Rob. | Clean | Rob. | Clean | Rob. | Clean | Rob. | Clean | Rob. | Clean | Rob. |
|---|---|---|---|---|---|---|---|---|---|---|---|---|---|---|---|---|---|---|---|---|---|---|---|
| General | ImageNet | 57.8 | 0.2 | 51.2 | 47.6 | 42.4 | **40.0** | 57.4 | 30.2 | 52.0 | 17.4 | 55.2 | 3.5 | 55.7 | 10.8 | **61.3** | 24.1 | 44.0 | 24.6 | 56.8 | 37.1 | -1.0 | +36.9 |
| | CIFAR10 | 86.1 | 0.6 | 50.3 | 50.2 | 47.2 | 46.9 | 76.7 | 35.1 | 84.9 | 27.2 | 86.5 | 4.2 | 84.4 | 41.8 | 86.0 | 33.7 | **87.6** | 43.3 | 86.1 | **58.1** | +0.0 | +57.5 |
| | CIFAR100 | 57.2 | 0.3 | 34.8 | 34.9 | 28.8 | 28.6 | 41.0 | 14.7 | 56.0 | 10.3 | **61.6** | 4.2 | 53.5 | 20.9 | 58.8 | 15.5 | 58.8 | 19.1 | 57.2 | **29.8** | +0.0 | +29.5 |
| | STL10 | 96.2 | 12.3 | 80.8 | 80.4 | 83.4 | 83.1 | 96.5 | 78.5 | 96.0 | 67.2 | 95.2 | 32.6 | 95.3 | 67.8 | **97.4** | **84.8** | 96.5 | 72.5 | 96.2 | 66.2 | +0.0 | +53.9 |
| | Caltech101 | 82.3 | 6.3 | 78.2 | 76.2 | 80.2 | 78.7 | **84.9** | 29.4 | 81.0 | 56.7 | 81.9 | 29.1 | 83.5 | 50.1 | 83.2 | 65.7 | 82.5 | 52.1 | 81.2 | 59.0 | -1.1 | +52.7 |
| | Caltech256 | 80.3 | 4.1 | 66.6 | 64.1 | 75.6 | 72.8 | 80.8 | 63.5 | 78.6 | 49.3 | 78.8 | 21.7 | **79.2** | 43.1 | 78.5 | 60.0 | 77.3 | 49.2 | **79.2** | 55.0 | -1.1 | +50.9 |
| Fine-G | OxfordPets | 84.2 | 0.2 | 68.6 | 64.3 | 67.5 | 61.6 | 84.2 | 60.2 | 74.6 | 25.2 | **84.2** | 6.4 | 83.6 | 19.9 | 81.7 | 24.8 | 82.2 | 30.7 | 83.9 | 49.4 | -0.3 | +49.2 |
| | Flowers102 | 62.7 | 0.8 | 21.5 | 22.6 | 44.3 | **42.7** | 59.2 | 35.2 | 57.8 | 25.0 | 59.8 | 6.2 | 60.9 | 14.2 | 61.5 | 14.2 | 61.9 | 26.4 | **62.6** | 32.0 | -0.1 | +31.2 |
| | Food101 | 80.4 | 0.1 | 24.1 | 22.1 | 38.3 | 36.6 | 82.1 | 57.1 | 71.0 | 20.3 | 79.7 | 3.6 | 78.5 | 11.4 | **79.9** | 46.0 | 72.7 | 36.5 | 79.4 | **57.7** | -1.0 | +57.6 |
| | StanfordCars | 59.9 | 0.0 | 26.1 | 20.5 | 60.3 | 52.1 | 60.2 | 28.1 | 44.4 | 5.1 | 50.5 | 3.2 | 58.1 | 4.9 | 51.2 | 26.6 | 49.0 | 16.3 | **59.6** | 25.4 | -0.3 | +25.4 |
| Scene | SUN397 | 61.9 | 0.8 | 32.2 | 32.9 | 42.2 | 40.1 | **63.1** | **54.2** | 59.3 | 21.3 | 58.1 | 6.1 | 61.7 | 13.2 | 62.2 | 18.0 | 53.1 | 31.2 | 61.1 | 48.3 | -0.8 | +47.5 |
| | Country211 | 15.7 | 0.0 | 2.2 | 2.7 | 4.1 | 4.4 | 14.3 | 5.0 | 13.1 | 1.2 | 12.1 | 0.0 | 12.9 | 0.8 | 16.3 | 1.2 | 12.6 | 3.7 | **15.5** | **12.1** | -0.2 | +12.1 |
| Domain | FGVCAircraft | 17.3 | 0.0 | 4.3 | 7.3 | 10.1 | 12.2 | 19.7 | 13.6 | 16.7 | 2.0 | 15.7 | 1.1 | 14.3 | 2.0 | **19.8** | 5.1 | 11.4 | 9.2 | 17.2 | **16.7** | -0.1 | +16.7 |
| | EuroSAT | 34.2 | 0.0 | 14.4 | 14.4 | 17.4 | 17.4 | 27.7 | 21.4 | 29.5 | 1.5 | **35.3** | 2.1 | 28.7 | 13.7 | 28.8 | 10.2 | 33.6 | 13.9 | 34.2 | **26.1** | +0.0 | +26.1 |
| | DTD | 43.3 | 1.7 | 25.8 | 24.9 | 32.3 | **31.6** | 39.6 | 32.6 | 39.9 | 18.0 | 40.3 | 11.4 | 40.9 | 15.6 | **44.2** | 23.2 | 42.9 | 22.1 | 42.4 | 25.8 | -0.9 | +24.1 |
| | PCAM | 48.9 | 24.8 | **55.7** | 55.6 | 51.0 | 50.8 | 55.4 | 41.2 | 48.7 | 48.2 | 49.0 | 39.8 | 48.8 | 48.3 | 54.5 | 38.7 | 48.8 | 44.2 | 49.1 | 50.3 | +0.2 | +25.5 |
| All | Avg. | 60.5 | 3.3 | 39.8 | 38.8 | 45.3 | **43.7** | 57.9 | 37.6 | 56.5 | 24.7 | 59.0 | 11.0 | 58.8 | 23.7 | 57.2 | 30.7 | 57.2 | 30.9 | **60.1** | 40.6 | -0.4 | +37.3 |

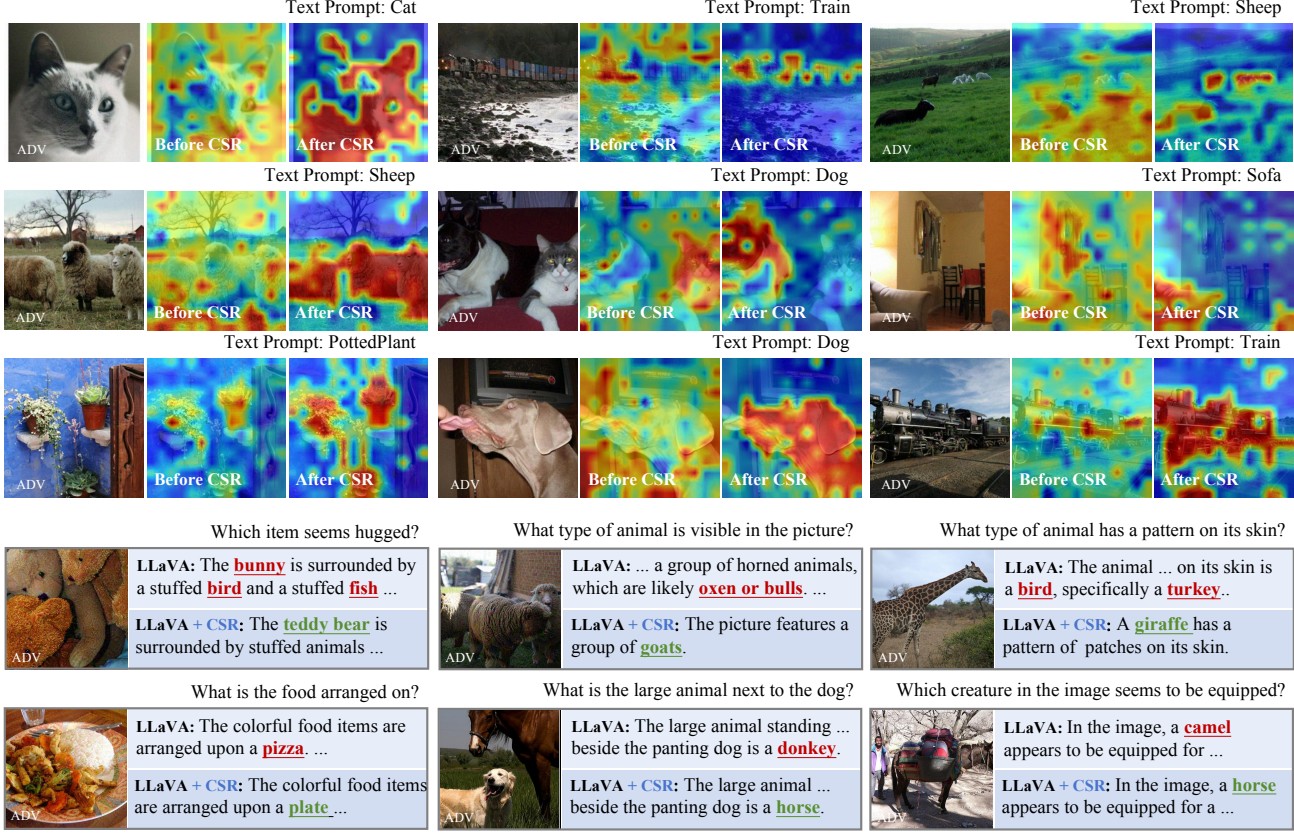

*Figure 7.* Additional Examples.

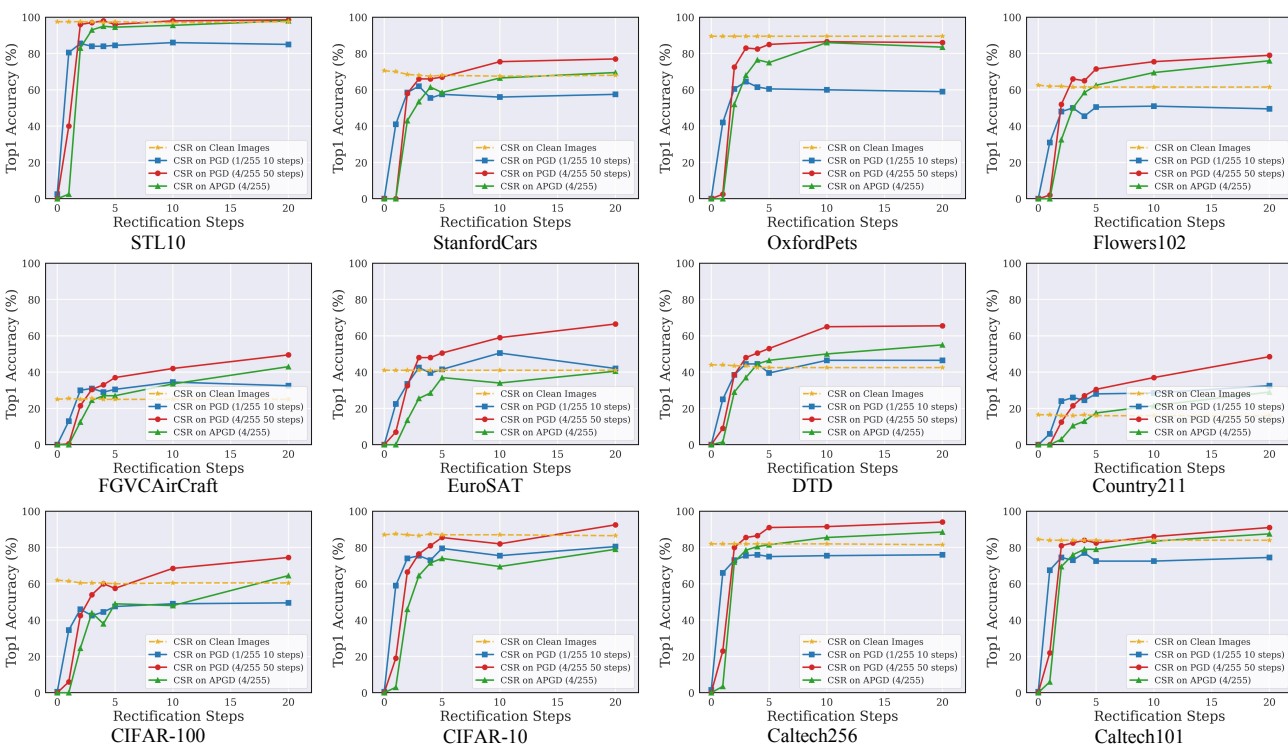

*Figure 8.* Ablation on the Rectification Steps.

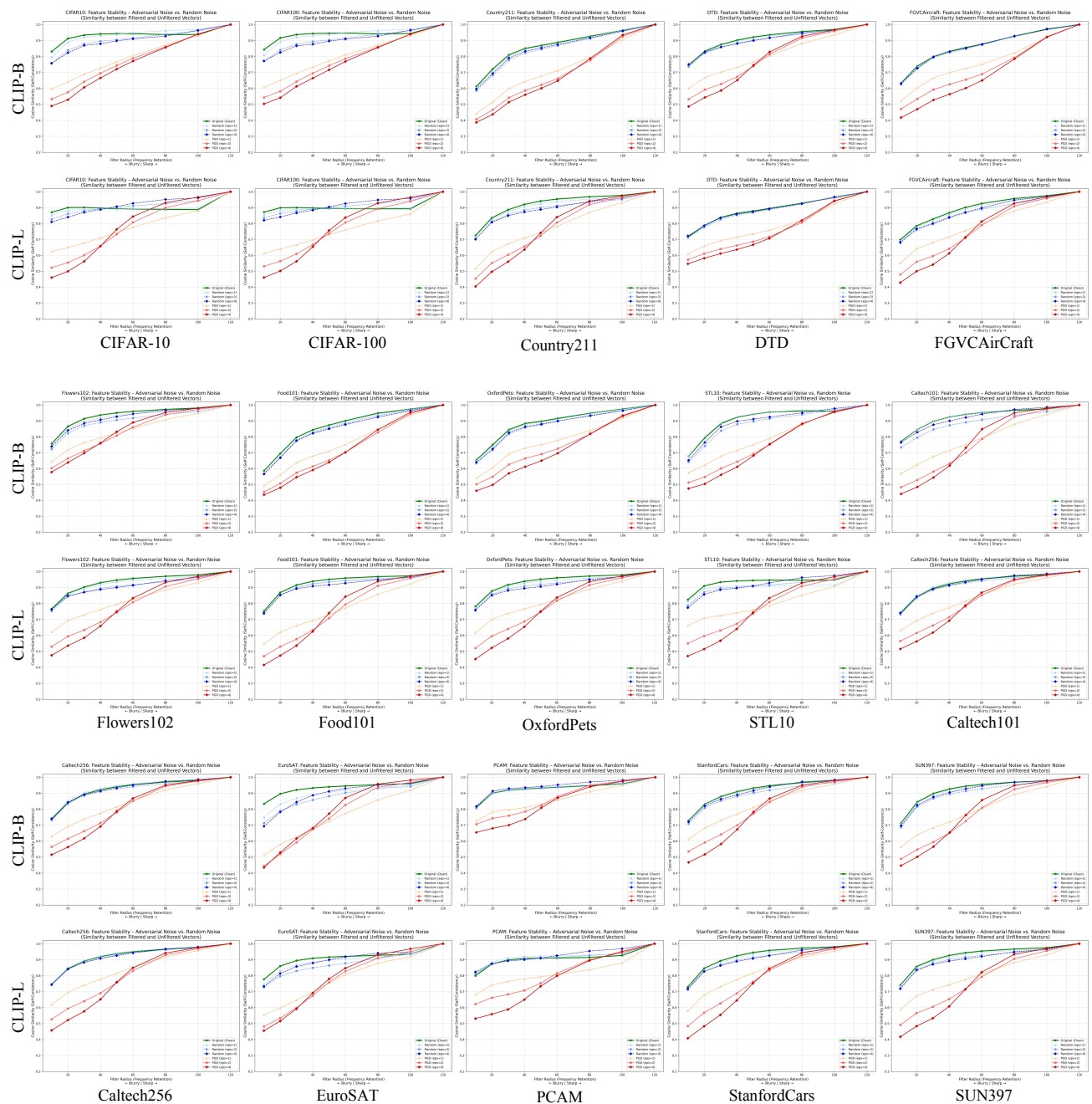

*Figure 9.* Spectral Analysis of CLIP Feature Consistency on widely used 15 Benchmark Datasets.

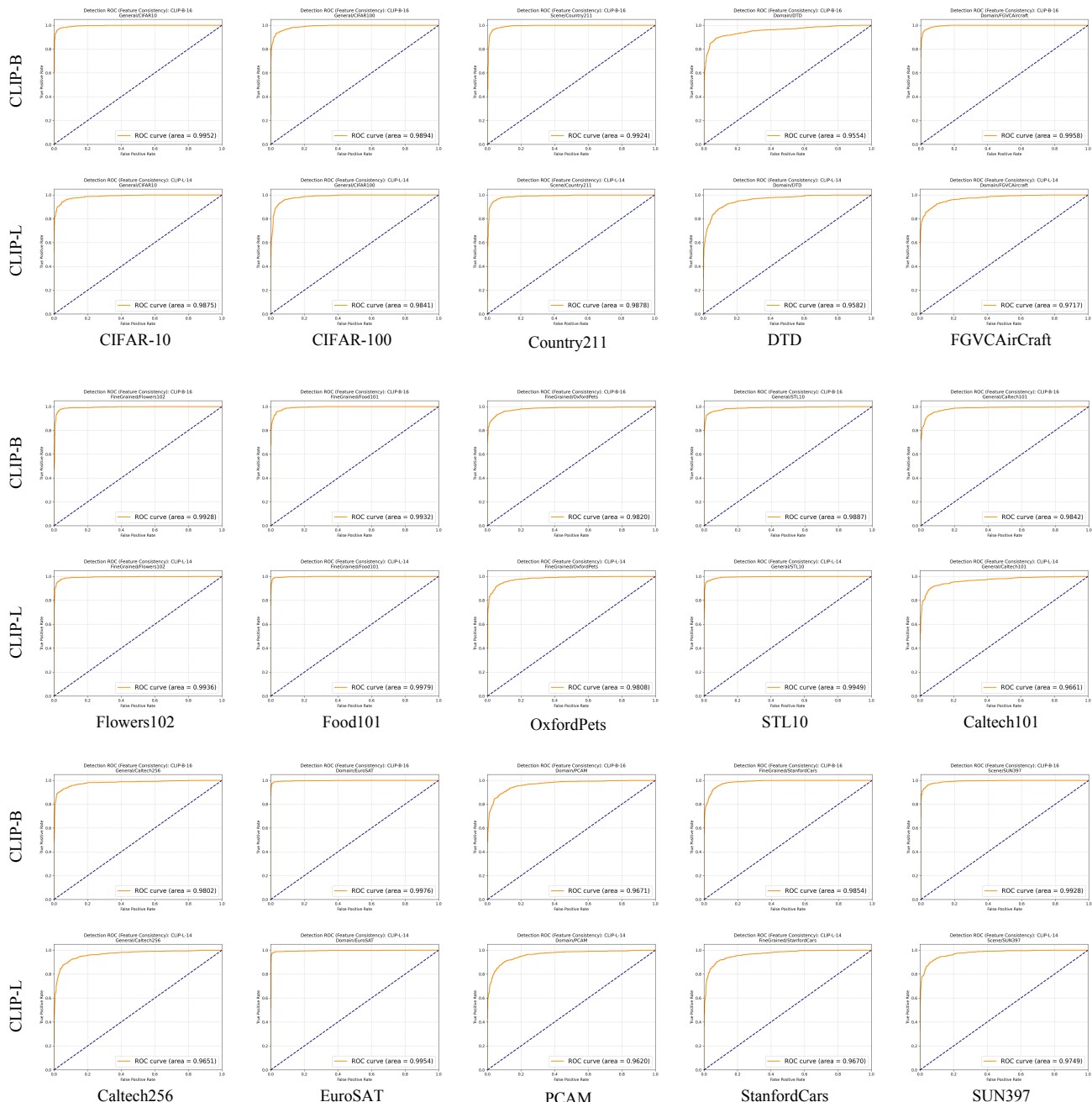

*Figure 10.* AUC curves for adversarial sample detection on 15 benchmark datasets.

