# Contrastive Spectral Rectification: Test-Time Defense towards Zero-shot Adversarial Robustness of CLIP

## Abstract

Vision-language models (VLMs) such as CLIP have demonstrated remarkable zero-shot generalization, yet remain highly vulnerable to adversarial examples (AEs). While test-time defenses are promising, existing methods fail to provide sufficient robustness against strong attacks and are often hampered by high inference latency and task-specific applicability. To address these limitations, we start by investigating the intrinsic properties of AEs, which reveals that AEs exhibit severe feature inconsistency under progressive frequency attenuation. We further attribute this to the model's inherent spectral bias. Leveraging this insight, we propose an efficient test-time defense named **C**ontrastive **S**pectral **R**ectification (CSR). CSR optimizes a rectification perturbation to realign the input with the natural manifold under a spectral-guided contrastive objective, which is applied input-adaptively. Extensive experiments across 16 classification benchmarks demonstrate that CSR outperforms the SOTA by an average of **18.1%** against strong AutoAttack with modest inference overhead. Furthermore, CSR exhibits broad applicability across diverse visual tasks. Code is available at https://anonymous.4open.science/r/CSR-3935.

## 1 Introduction

Large-scale pre-trained Vision-Language Models (VLMs), notably CLIP (Radford et al., 2021), have revolutionized multimodal representation learning with their remarkable zero-shot generalization (Shen et al., 2022; Jing et al., 2024; Awais et al., 2025). Despite this success, CLIP models remain highly vulnerable to adversarial examples (Zhao et al., 2024; He et al., 2025). Imperceptible perturbations can easily deceive the model, undermining its reliability in safety-critical open-world applications (Chen et al., 2025a).

[1]Anonymous Institution, Anonymous City, Anonymous Region, Anonymous Country. Correspondence to: Anonymous Author <anon.email@domain.com>.

Preliminary work. Under review by the International Conference on Machine Learning (ICML). Do not distribute.

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

1: $\mathbf{z} \leftarrow f(\mathbf{x}), \mathbf{z}_{low} \leftarrow f(G_r(\mathbf{x}))$
2: $s \leftarrow \cos(\mathbf{z}, \mathbf{z}_{low})$ ▷ *(i) Adversarial Examples Detection*
3: **if** $s \geq \tau$ **then**
4:     **return x**           ▷ *Consistent → Benign Image*
5: **else**
6:     $\delta \leftarrow \mathbf{0}, \mathbf{x}' \leftarrow \mathbf{x} + \delta, \mathbf{x}^* \leftarrow \mathbf{x}, \mathcal{L}_{best} \leftarrow -\infty$
7:     **for** $t = 1$ **to** $N$ **do**
8:         ▷ *(ii) Few-step Contrastive Spectral Rectification*
9:         $\mathbf{z}' \leftarrow f(\mathbf{x}')$
10:        $\mathcal{L}_{rec} \leftarrow \cos(\mathbf{z}', \mathbf{z}_{tgt}) - \cos(\mathbf{z}', \mathbf{z}_{adv})$
11:        **if** $\mathcal{L}_{rec} > \mathcal{L}_{best}$ **then**
12:           $\mathcal{L}_{best} \leftarrow \mathcal{L}, \mathbf{x}^* \leftarrow \mathbf{x}'$   ▷ *(iii) Greedy Selection*
13:        **end if**
14:        $\mathbf{x}' \leftarrow \mathbf{x}' + clip(\delta + \alpha \cdot \text{sign}(\nabla_{\mathbf{x}'}\mathcal{L}), -\epsilon, \epsilon)$
15:     **end for**
16: **end if**
17: **return** $\mathbf{x}^*$       ▷ *Return Rectified Reliable Image*

---

significant feature collapse. We quantify this divergence using the Cosine Similarity between the embeddings:

$$\mathcal{C}(\mathbf{x}) = \frac{f(\mathbf{x})^\top f(\mathbf{x}_{low})}{\|f(\mathbf{x})\|_2 \|f(\mathbf{x}_{low})\|_2}. \quad (5)$$

A high $\mathcal{C}(\mathbf{x})$ indicates that the semantic content is robustly anchored in the low-frequency band, characteristic of BEs. Conversely, a low score reveals a reliance on fragile mid-to-high frequencies—a signature of AEs. We classify an input as adversarial if $\mathcal{C}(\mathbf{x}) < \tau$, where $\tau$ is a detection threshold to balance detection sensitivity and the false positive rate. Detection ROC curves are provided in Appendix D.

### 4.2 Contrastive Spectral Rectification

While filtering is effective for detection, naive application of low-pass filters risks over-smoothing, eroding fine-grained details essential for zero-shot recognition. To overcome this, we propose Contrastive Spectral Rectification. Instead of the passive suppression of mid-to-high frequency components, CSR formulates an active test-time optimization that reconstructs a reliable input $\mathbf{x}^*$ from the adversarial query.

We freeze the model parameters and optimize a rectification perturbation $\boldsymbol{\delta}$ to get the rectified sample $\mathbf{x}' = \mathbf{x} + \boldsymbol{\delta}$. The objective function, $\mathcal{L}_{rec}$, is designed as follows:

$$\mathcal{L}_{rec}(\boldsymbol{\delta}) = \underbrace{\text{sim}(f(\mathbf{x} + \boldsymbol{\delta}), f(\mathbf{x}_{low}))}_{\text{Attraction Term}} - \lambda \cdot \underbrace{\text{sim}(f(\mathbf{x} + \boldsymbol{\delta}), f(\mathbf{x}))}_{\text{Repulsion Term}}, \quad (6)$$

where $\text{sim}(\cdot, \cdot)$ denotes cosine similarity and $\lambda$ is a hyperparameter to balance the two terms.