# OpenReview forum: "Contrastive Spectral Rectification: Test-Time Defense towards Zero-shot Adversarial Robustness of CLIP"
_ICML.cc/2026/Conference — ICML 2026 regular_

### Official Review · Reviewer_3gNb · 2026-02-23

**Soundness:** 3
**Presentation:** 3
**Significance:** 3
**Originality:** 2
**Overall Recommendation:** 5
**Confidence:** 3

**Summary:**

The Authors introduce CSR - an inference-time defense mechanism that realign potential adversarial samples into its low-frequency features’  ’natural’ manifold. Based on the investigation of adversarial examples (AEs) for CLIP, they discover that CLIP is vulnerable against AEs mostly within mid-to-high frequency components. They propose a defense that detect and purify AEs during inference with the use of low-frequency features as attraction anchor, and high-frequency features as repulsion anchor.

**Compliance With Llm Reviewing Policy:**

Affirmed.

**Final Justification:**

I maintain my previous Accept score after clarifications during the rebuttal. I see this work as an interesting contribution.

I believe that in the revised version the authors may clarify raised issues and give a broader view of frequency-patterns vulnerabilities.

While I disagree with the authors on some non-critical points, it may be the limitations of the rebuttal communication that makes it harder to understand each other and those points could be discussed naturally e.g. during a poster session.

I am open to discuss this contribution further with other reviewers and possibly adjust my current evaluation.

**Key Questions For Authors:**

1. Is mid-to-high frequency vulnerability the same for all tested visual backbones? Wider comparison with different metrics would be helpful.
2. Figure 3: There is a need for ablation in the reverse direction to strengthen your claims -> starting from only high-pass filter and going towards low frequencies. It is possible that mid-to-low frequencies would work as adversarial deceptive signals as well, which would weaken method motivation.
3. What are the limitations of the proposed method?

**Limitations:**

Not mentioned in the main part.

**Strengths And Weaknesses:**

**Strengths**:
It is a good work. The work is well-written, easy to follow and an interesting read. To my best knowledge mentioned related work is relevant and the presented context strengthens the motivation of the current work (presentation). The main part contain clearly described and diverse experimental results, all basic ablations needed to understand the parts and importance of the final solution (soundness).  It is approaching a relevant problem, and shows interesting idea for a partial solution that can be used effectively (significance). The combination of previously existing approaches is well-motivated and clearly presented with the good test-time defense results shown (originality).

**Weaknesses**:
*Line 200-201*: a bit overstatement, since it is true only for tested vision encoder pretraining, given that AEs frequency is a more complex issue connected with the pretraining data [1].  I think it needs a clarification in the light of the above work, but maybe it is understandable in the current work context (soundness).

 *Line 245*: lacks what is the base work for the experimental setup - no citation present (presentation).

[1] A FREQUENCY PERSPECTIVE OF ADVERSARIAL ROBUSTNESS, 2021

---

> ### Author Rebuttal · Authors · 2026-03-30
>
> We sincerely thank the reviewer for recognizing CSR as a **well-motivated and interesting** defense with **strong empirical validation on an important problem**. We address the concern as follows.
>
> ---
>
> > Q1: *Line 200-201*: a bit overstatement, since it is true only for tested vision encoder pretraining, given that AEs frequency is a more complex issue connected with the pretraining data [1] ...
>
> We agree that frequency preference may be influenced by both the pretraining data and the model architecture. We will **clarify this point more carefully** in the revision and cite the relevant literature accordingly. In our setting, CLIP appears to develop a certain intrinsic frequency preference under large-scale pretraining and adversarial examples further exhibit a bias toward mid-to-high frequency.
>
> ---
>
> > Q2: *Line 245*: lacks what is the base work for the experimental setup - no citation present.
>
> Thank you for pointing out this omission. Our experimental setup **strictly follows the settings used in the relevant baselines**, including TTC [CVPR'25], R-TPT [CVPR'25], and FARE [ICML'24 Oral]. We will add the corresponding citations in the revision for clarity.
>
> ---
>
> > Q3: Is mid-to-high frequency vulnerability the same for all tested visual backbones? Wider comparison with different metrics would be helpful.
>
> Thank you for the helpful suggestion.
>
> In the main paper, we already observe a similar mid-to-high frequency vulnerability across multiple visual backbones, including CLIP-B/32, CLIP-B/16, CLIP-L/14, and CLIP-L/14@336 used in LLaVA. We further observe the same tendency for Qwen’s self-trained visual encoder (see our response to Reviewer pvFZ, Q4). This suggests that such **spectral vulnerability is common across large-scale pretrained vision-language models**. CSR leverages this shared property to improve robustness effectively across models.
>
> To further address this question, we provide a quantitative metric: the gradient distribution ratio across frequency bands. As shown in Table I, for all tested models, around 90% of gradients are concentrated in the mid-to-high frequency bands. This further supports that adversarial vulnerability is predominantly concentrated in the mid-to-high frequency range across backbones, a property that is leveraged by CSR.
>
> **Table I**. Gradient distribution ratio (%) across frequency bands on ImageNet classification task.
>
> |                       | CLIP-B/32 | CLIP-B/16 | CLIP-L/14 | CLIP-L/14@336 |
> | --------------------- | --------- | --------- | --------- | ------------- |
> | Low frequency         | 9.79      | 11.02     | 10.74     | 5.23          |
> | Mid-to-high frequency | 90.21     | 88.98     | 89.26     | 94.77         |
>
> ---
>
> > Q4: Figure 3: There is a need for ablation in the reverse direction to strengthen your claims -> starting from only high-pass filter and going towards low frequencies ...
>
> Thank you for this thoughtful suggestion. To further examine this point, we perform the reverse ablation by applying high-pass filtering and going towards low frequencies. The results are shown in Table II.
>
> These results show that under high-pass filtering, CLIP is almost unable to perform classification: even clean-image accuracy is almost zero. This indicates that the **low-frequency component carries indispensable core semantics**, while adversarial examples can exploit CLIP’s mid-to-high-frequency vulnerability to mislead semantic prediction. Importantly, such mid-to-high-frequency deception does not appear to work in isolation from the image’s low-frequency content, which is also consistent with the fact that adversarial examples are generated based on the original image.
>
> Overall, **this reverse ablation further supports our findings** in Sec. 3.2: CSR leverages CLIP’s intrinsic frequency characteristics to achieve effective test-time adversarial defense.
>
> **Table II.** ImageNet classification accuracy (%) of CLIP-B/16 under different high-pass filters.
>
> |            | r=10 | r=20 | r=30 | r=40 | r=50 |
> | ---------- | ---- | ---- | ---- | ---- | ---- |
> | Clean      | 0.0  | 0.0  | 0.2  | 0.0  | 0.2  |
> | APGD 1/255 | 0.2  | 0.2  | 0.2  | 0.2  | 0.2  |
> | APGD 2/255 | 0.0  | 0.2  | 0.2  | 0.2  | 0.2  |
> | APGD 4/255 | 0.0  | 0.2  | 0.2  | 0.4  | 0.2  |
>
> ---
>
> > Q5: What are the limitations of the proposed method?
>
> Thank you for the valuable question. While CSR is an efficient and broadly effective test-time defense, its advantage—similar to most existing methods—can become less pronounced under global color attacks (in our response to Reviewer kfam (Q3)), which are rarely considered in standard evaluation settings. Nevertheless, CSR still provides improvement over standard CLIP under such attacks, and we view this as a useful boundary case that helps further clarify the scope of current test-time defenses and motivates future study.

---

> > ### Author Rebuttal · Reviewer_3gNb · 2026-04-02
> >
> > Thank you for the answers. Regarding my questions for ablations Figure 3 (you mention it as Q4): 1) Table II is not what I asked for. Figure 3 shows features disparity not acc. 2) Is there a counterpart for this Table II in the main paper for low-frequencies?
> >
> > Comment of method generality: It may be true that CLIP pretraining has a strong spectral bias towards low-frequencies, but at the same time it is not general, the pretraining done differently may have adversarial vulnerabilities in low-frequencies as well.
> >
> > While it is possible that some CLIP pretraining may bias AEs towards mid-high frequencies (the core motivation for your method), it is **highly improbable** that an attack on low-frequencies is not possible (given previous literature, the scale of datasets used for CLIP pretraining and adversarial vulnerabilities of ImageNet images). The attack would slightly alter low-frequencies, and (I assume) your defense would fail in that context. Could it be seen as a limitation? It is vital to clarify this point now and in the final revised version of this work, not to confuse or misinterpret (overclaim) obtained results and clearly see scope, generality and the limitations of the proposed method.

---

> > > ### Author Response · Authors · 2026-04-03
> > >
> > > We sincerely thank the reviewer for their feedback. We address the follow-up questions as follows.
> > >
> > > ---
> > >
> > > > Q1: Regarding my questions for ablations Figure 3 (you mention it as Q4): 1) Table II is not what I asked for. Figure 3 shows features disparity not acc. 2) Is there a counterpart for this Table II in the main paper for low-frequencies?
> > >
> > > Thank you for pointing this out.
> > >
> > > We provide additional results below to further clarify this issue, and **these results are consistent with the findings in the main paper**.
> > >
> > > (1) The corresponding results for progressively applying low-pass filtering to standard CLIP on ImageNet classification (Figure 3 in the main paper) are shown below:
> > >
> > > | Type       | 70       | 50   | 40   | 30   | 20       | 10   | 5    |
> > > | ---------- | -------- | ---- | ---- | ---- | -------- | ---- | ---- |
> > > | Clean      | **63.0** | 61.2 | 59.4 | 57.2 | 49.0     | 31.8 | 6.6  |
> > > | APGD_2/255 | 0.2      | 8.8  | 23.4 | 35.4 | **38.3** | 27.6 | 5.8  |
> > > | APGD_4/255 | 0.0      | 2.4  | 10.8 | 29.2 | **35.1** | 27.0 | 5.2  |
> > >
> > > (2) The feature visualization for the reverse ablation (high-pass filtering) mentioned by the reviewer is provided at the anonymized link: https://anonymous.4open.science/r/CSR-3935/clip_b_16_imagenet_highpass_cosine.png.
> > >
> > > ---
> > >
> > > > Q2: Comment of method generality: It may be true that CLIP pretraining has a strong spectral bias towards low-frequencies, but at the same time it is not general, the pretraining done differently may have adversarial vulnerabilities in low-frequencies as well.
> > >
> > > Thank you for the question.
> > >
> > > While we agree that frequency preferences are not identical across different models and do exhibit certain variations, we would like to clarify that **they still share common patterns, especially among large-scale pretrained VLMs**. Our reasoning is as follows:
> > >
> > > **(1) Empirically, we observe consistent frequency-level commonalities across large vision-language models, and CSR consistently improves their robustness.** As shown in our additional results, representative models—including CLIP-B/32, CLIP-B/16, CLIP-L/14, as well as LLaVA (with a CLIP-L/14@336 backbone) and Qwen (with a self-trained ViT encoder)—exhibit similar frequency preferences in their decision processes, with mid-to-high frequency components being more vulnerable to adversarial perturbations.
> > >
> > > **(2)** **We believe these commonalities are closely related to the large-scale pretrained vision-language modeling paradigm.** Specifically: **(i)** These models are trained on massive datasets, such as LAION-400M, LAION-5B, and DataComp (1.28B). These datasets are roughly 300–4,000× larger than ImageNet and broadly cover the dominant visual content on the internet. **(ii)** The core training paradigm of vision-language models is to align image and text modalities for unified visual-semantic understanding. **(iii)** Architecturally, these models predominantly adopt Transformer-based designs to enable large-scale learning. These similarities in training data, training objectives, and model architectures **may lead to common frequency characteristics**. This, in turn, suggests that CSR has potential to generalize across diverse large-scale pretrained VLMs.
> > >
> > > ---
> > >
> > > > Q3: ... it is highly improbable that an attack on low-frequencies is not possible ... The attack would slightly alter low-frequencies, and (I assume) your defense would fail in that context ... It is vital to clarify this point now and in the final revised version of this work...
> > >
> > > Thank you for the insightful question.
> > >
> > > In the main paper, we state that attacks constrained to low-frequency components are more difficult to optimize, but not impossible. We understand the reviewer’s concern that such **low-frequency attacks** may potentially undermine our defense. This corresponds to a strong adaptive attack specifically targeting CSR and therefore provides a stringent test of its robustness. This concern is closely related to Reviewer pvFZ (Q1) and Reviewer kfam (Q1), and **we have therefore conducted rigorous evaluations** under adaptive attack settings. Due to space constraints, we refer the reviewer to our response to Reviewer pvFZ (Q1) for detailed results on fully adaptive attacks.
> > >
> > > In brief, although CSR exhibits some performance degradation in this setting, **it remains highly effective** and consistently outperforms TTC [CVPR’25]. On ImageNet under adaptive attack, CSR improves the robustness of CLIP from 14.1% to 46.3%, substantially exceeding TTC (29.1%). These results indicate that **our defense does not fail and maintains a clear advantage over the baseline**. The effectiveness of CSR in this setting has also been acknowledged by Reviewer kfam.
> > >
> > > ---
> > >
> > > We sincerely thank the reviewer for their time and effort. We hope the above responses address the concerns, and we would be glad to clarify any remaining points if needed.

---

### Official Review · Reviewer_e59U · 2026-02-27

**Soundness:** 3
**Presentation:** 3
**Significance:** 3
**Originality:** 2
**Overall Recommendation:** 3
**Confidence:** 4

**Summary:**

This paper studies the role of frequency components in adversarial robustness and proposes a frequency-aware perturbation strategy to improve robustness under constrained noise budgets. The method introduces a structured perturbation mechanism that operates in the spectral domain and aims to better align adversarial perturbations with robustness-relevant frequency bands.

**Compliance With Llm Reviewing Policy:**

Affirmed.

**Final Justification:**

While some of my concerns have been addressed, I still have the following reservations that are not fully addressed:
- In Contribution 1, the paper claims: "We uncover a spectral fragility in adversarial examples and trace this vulnerability to CLIP’s spectral bias and sensitivity to mid-to-high frequency components." However, there is a growing body of research indicating that low-frequency features tend to be more robust than high-frequency ones empirically and theoretically. The described "a spectral fragility in adversarial examples" and "sensitivity to mid-to-high frequency components" thus appear to be a general phenomenon, rather than a finding specific to the setting of this work.
- The rectification budget, which is an important hyper-parameter of the proposed method, is fixed at 4/255 in the original paper. Although the authors provide an additional hyper-parameter analysis in Table II, the results suggest that 4/255 is not the optimal setting. This may indicate that the hyper-parameters in this study require more systematic tuning.
Therefore, I maintain my initial score.

**Key Questions For Authors:**

See weakness.

**Limitations:**

yes

**Strengths And Weaknesses:**

Strengths

- The paper explores adversarial robustness from a frequency-domain perspective, which provides an alternative viewpoint beyond purely spatial perturbation design.
- The proposed method is simple and can be incorporated into existing training pipelines with limited architectural modification.

Weaknesses

1. There has been a growing body of work studying the relationship between frequency and robustness, many of which suggest that low-frequency features are more robust than high-frequency ones. For example:
   - Garg S, et al. A spectral view of adversarially robust features. NeurIPS 2018.
   - Wang H, et al. High-frequency component helps explain the generalization of convolutional neural networks. CVPR 2020.
   - Bu Q, et al . Towards building more robust models with frequency bias. ICCV 2023.
   The paper would benefit from a clearer comparison with these works and a more explicit discussion of what distinguishes the proposed method.
2. Some symbols in Algorithm 1 appear inconsistent with those in the main text, which may cause confusion when following the implementation details.
3. The adversarial perturbation budget used for evaluation is set to 1/255 unless otherwise specified. This is significantly smaller than widely adopted settings (e.g., 8/255 on CIFAR-10/100), making it difficult to fairly assess robustness improvements.
4. In Tables 1 and 2, there are cases where the accuracy under attack  is higher than the clean accuracy. This counter-intuitive observation would benefit from further explanation.
5. The perturbation budget of the proposed method is fixed at 4/255, but no sensitivity analysis is provided. A discussion or ablation study on this hyperparameter would help better understand its effect.

---

> ### Author Rebuttal · Authors · 2026-03-30
>
> We sincerely thank the reviewer for the valuable feedback. We respond to the concerns as follows.
>
> ---
>
> > Q1: ... The paper would benefit from a clearer comparison with these works and a more explicit discussion of what distinguishes the proposed method.
>
> Thank you for pointing us to these relevant works. We agree that they are related in studying frequency and adversarial robustness, and we will clarify the comparison in the revision.
>
> Our method **differs** from them in several concrete ways.
>
> (1) Garg et al. [NeurIPS’18] and Bu et al. [ICCV’23] improve robustness through training-time learning or model-side design, whereas CSR is a training-free, plug-and-play test-time defense that requires neither retraining nor architectural modification.
>
> (2) Wang et al. [CVPR’20] mainly analyzes frequency effects in CNNs, whereas CSR revisits this question in pretrained VLMs and further connects frequency properties to feature stability through both empirical and theoretical analyses (Sec. 3.2). Moreover, CSR introduces a practical test-time defense based on input-adaptive contrastive rectification, rather than analysis alone.
>
> ---
>
> > Q2: Some symbols in Algorithm 1 appear inconsistent with those in the main text ...
>
> Thank you for pointing this out. In our paper, both $s$ and $C(\cdot)$ denote cosine similarity. To avoid confusion, we will **unify this notation** as $s$ in the revision and carefully correct other symbol inconsistencies throughout the paper as well. The anonymized link we previously provided in the main paper contains the full reproduction code and may help clarify the implementation details.
>
> ---
>
> > Q3: The adversarial perturbation ... is significantly smaller than widely adopted settings (e.g., 8/255 on CIFAR-10/100) ...
>
> The perturbation budgets used in the main paper (1/255 and 4/255) **strictly follow the settings of the most relevant baselines** for CLIP test-time defense, including TTC [CVPR'25], R-TPT [CVPR'25], and FARE [ICML'24 Oral], thereby ensuring a fair comparison. We agree that evaluation under larger perturbation budgets is also important for further assessing the robustness of CSR. In response, we provide the following results.
>
> (1)  In the main paper, our experiments on VQA and Captioning tasks already use a larger budget of 16/255 (Table 7 and Fig. 6), where CSR remains effective.
>
> (2) We further add results under PGD with 8/255. As shown in Table I, **CSR remains effective** under this stronger setting, and increasing the number of rectification steps $N$ further improves performance.
>
> **Table I**. Classification accuracy (%) of CLIP-B/16 under PGD attack with perturbation budget 8/255.
>
> |                     | ImageNet | Flowers102 | SUN397 | PCAM |
> | ------------------- | -------- | ---------- | ------ | ---- |
> | Standard CLIP       | 0.0      | 0.0        | 0.0    | 0.0  |
> | TTC [CVPR'25] (N=3) | 1.6      | 0.5        | 1.1    | 2.1  |
> | TTC [CVPR'25] (N=5) | 4.1      | 2.7        | 3.2    | 6.1  |
> | CSR (N=3)           | 38.2     | 41.8       | 39.6   | 46.6 |
> | CSR (N=5)           | 52.9     | 58.6       | 54.2   | 52.4 |
>
> ---
>
> > Q4: In Tables 1 and 2, there are cases where the accuracy under attack is higher than the clean accuracy. This counter-intuitive observation would benefit from further explanation.
>
> Thank you for the thoughtful observation. This phenomenon is also observed in [1]. As discussed in our ablation and **consistent with [1, Sec. 4.2]**, "adversarial examples generated with ground-truth labels may implicitly encode directional priors about the true decision boundary". CSR exploits this signal through contrastive rectification, estimating a defense direction and shifting features along it to recover more robust representations. Moreover, CSR remains effective when adversarial examples are generated without ground-truth labels, as shown in Tables 4 and 7 of the main paper.
>
> [1] Liu, L. et al. "Adversarial Attacks Already Tell the Answer: Directional Bias-Guided Test-time Defense for Vision-Language Models." ICLR, 2026.
>
> ---
>
> > Q5: The perturbation budget of the proposed method is fixed at 4/255 ...
>
> Thank you for pointing this out. The rectification budget is fixed to 4/255, **following** TTC [CVPR'25]. To further examine the sensitivity to this hyperparameter, we conduct an ablation on ImageNet, with the rectification step number fixed to N=3.
>
> **Table II.** Classification accuracy (%) of CLIP-B/16 under APGD (2/255) with different rectification budgets (columns) and rectification step sizes (rows).
>
> |       | 2/255 | 4/255 | 8/255 |
> | ----- | ----- | ----- | ----- |
> | 1/255 | 55.2  | 61.9  | 61.9  |
> | 2/255 | 55.1  | 65.1  | 66.4  |
> | 4/255 | 51.5  | 62.1  | 68.3  |
>
> These results show that a larger rectification budget generally leads to stronger robustness. While overly small budgets or mismatched step-size/budget combinations may weaken CSR, the method **remains effective across a broad range of parameter choices**.

---

> > ### Author Rebuttal · Reviewer_e59U · 2026-04-01
> >
> > Thank you for your response. While some of my concerns have been addressed, I still have the following reservations:
> >
> > 1. In Contribution 1, the paper claims: "We uncover a spectral fragility in adversarial examples and trace this vulnerability to CLIP’s spectral bias and sensitivity to mid-to-high frequency components." However, there is a growing body of research indicating that low-frequency features tend to be more robust than high-frequency ones. The described "a spectral fragility in adversarial examples" and "sensitivity to mid-to-high frequency components" thus appear to be a general phenomenon, rather than a finding specific to the setting of this work.
> >
> > 2. The rectification budget, which is an important hyper-parameter of the proposed method, is fixed at 4/255 in the original paper. Although the authors provide an additional hyper-parameter analysis in Table II, the results suggest that 4/255 is not the optimal setting. This may indicate that the hyper-parameters in this study require more systematic tuning.

---

> > > ### Author Response · Authors · 2026-04-01
> > >
> > > We sincerely thank the reviewer for their time and effort during the review process. The following is a detailed response to the reviewer’s further questions.
> > >
> > > ---
> > >
> > > > Q1: In Contribution 1... there is a growing body of research indicating that low-frequency features tend to be more robust than high-frequency ones. The described "a spectral fragility in adversarial examples" and "sensitivity to mid-to-high frequency components" thus appear to be a general phenomenon, rather than a finding specific to the setting of this work.
> > >
> > > We fully acknowledge prior work, but we would like to respectfully clarify the distinction between our contribution and existing studies.
> > >
> > > (1) **From supervised CNNs/ViTs to contrastively pretrained CLIP.** Although models may share some similar frequency characteristics in many cases, differences in architecture [1] and training type [2] can influence frequency preferences. To the best of our knowledge, we are the first to identify and systematically study the frequency properties of adversarial examples on large pretrained VLMs, thereby extending prior work.
> > >
> > > (2) **From observation to mechanism.** We go beyond empirical observation by providing a fine-grained analysis, offering **a new perspective** through frequency-aware investigation of model decision-making and band-wise analysis of feature consistency. Furthermore, we **theoretically show** that the vulnerability and reliance of models on mid-to-high frequency components make low-frequency–constrained adversarial optimization significantly more difficult, suggesting that such vulnerability is inherent. Our empirical analyses and theoretical results complement and extend existing studies on robustness and frequency.
> > >
> > > (3) **Most importantly, we translate these findings and analyses into a novel and effective defense strategy**. CSR is **the first** test-time defense to leverage this frequency-robustness relationship via contrastive spectral guidance for both adversarial example detection and rectification. **Extensive experiments** on 16 datasets demonstrate that CSR outperforms prior state-of-the-art methods by an average of **↑18.1%** under strong AutoAttack, while reducing the average inference overhead **from 177.33 ms to 15.17 ms**. In addition, we show that, whereas most existing test-time defenses are limited to classification tasks, CSR generalizes effectively across segmentation (**↑16.7%**), visual question answering (**↑43%**), and image captioning (**↑43%**), and further exhibits strong potential for scaling to larger vision-language models (e.g., improving robustness by **↑43%** on LLaVA-1.5 and **↑39%** on Qwen2.5-VL).
> > >
> > > **In summary**, we extend the current understanding of frequency and robustness, provide comprehensive empirical and theoretic alanalyses, and build upon these insights to propose a novel, efficient, effective, and broadly applicable test-time defense.
> > >
> > > [1] Yutong Bai et al. Are Transformers More Robust Than CNNs? NeurIPS 2021
> > >
> > > [2] Maiya et al., A Frequency Perspective of Adversarial Robustness, arXiv 2021.
> > >
> > > ---
> > >
> > > > Q2: The rectification budget ... is fixed at 4/255 in the original paper. Although the authors provide an additional hyper-parameter analysis in Table II, the results suggest that 4/255 is not the optimal setting. This may indicate that the hyper-parameters in this study require more systematic tuning.
> > >
> > > **We thank the reviewer for this observation. We respond as follows:**
> > >
> > > **(1)** We adopt this setting to **follow the most closely related baseline (TTC, CVPR’25) for fair comparison**, rather than tuning it for optimality. As shown in the ablation study (Table II), CSR is not sensitive to precise hyperparameter choices and **remains effective across a broad range of settings**.
> > >
> > > **(2)** Although CSR has the potential to achieve further robustness gains under more fine-grained tuning, the current configuration with a perturbation budget of 4/255 **already yields strong performance**. **Compared with prior state-of-the-art methods**, CSR improves robustness by an average of ↑18.1% on 16 classification datasets under strong AutoAttack, achieves a ↑16.7% gain on segmentation tasks, and further improves robustness by ↑43% on LLaVA and ↑39% on Qwen in visual question answering.
> > >
> > > ---
> > >
> > > We have carefully responded to each of the questions, and we hope these clarifications address the reviewer’s concerns. We welcome any further questions or comments.

---

### Official Review · Reviewer_kfam · 2026-03-11

**Soundness:** 3
**Presentation:** 3
**Significance:** 3
**Originality:** 2
**Overall Recommendation:** 5
**Confidence:** 4

**Summary:**

This paper proposes Contrastive Spectral Rectification (CSR), a test-time defense framework designed to improve the adversarial robustness of CLIP models. The key observation is that adversarial examples tend to rely on high-frequency components, while low-pass filtered images remain close to the clean semantic manifold in the CLIP feature space. Based on this observation, the proposed method introduces a spectral contrastive rectification mechanism that attracts adversarial features toward a low-frequency anchor while repelling them from the adversarial feature region.

The method first uses a similarity test between the original image and its low-pass filtered version to detect potential adversarial inputs. When the similarity falls below a threshold, CSR performs a rectification optimization that adjusts the input to align its feature representation with the low-pass anchor. Importantly, this process operates entirely at test time and does not require retraining the model.

Experiments across a wide range of vision tasks—including classification, segmentation, captioning, and VQA—demonstrate consistent robustness improvements. In particular, CSR significantly improves performance under strong attacks such as AutoAttack, outperforming prior test-time defense methods.

**Compliance With Llm Reviewing Policy:**

Affirmed.

**Final Justification:**

The rebuttal adequately addressed my concerns regarding adaptive attack robustness and threshold stability. The additional evaluations further support the reliability and generality of the proposed mechanism. While global color transformations remain challenging, they fall outside the primary threat model and do not undermine the practical effectiveness of CSR. Overall, the method is technically sound and empirically well-supported, and I revise my assessment positively.

**Key Questions For Authors:**

The following questions correspond to the main concerns described in the weaknesses above.

1. **Adaptive attack robustness**:
The CSR recovery algorithm is composed of differentiable operations. Have the authors evaluated the defense under fully adaptive attacks where the attacker explicitly optimizes perturbations through the CSR pipeline?

2. **Threshold sensitivity**:
The detection mechanism relies on a cosine similarity threshold between the input image and its low-pass filtered version. How sensitive is the method to the choice of this threshold, and how does varying the threshold affect the final robust accuracy and false positive rate?

3. **Spectral assumption boundary cases**:
CSR assumes that adversarial perturbations mainly reside in mid- and high-frequency components. How does the method behave under attacks that may violate this assumption, such as localized patch attacks or global color manipulations?

**Limitations:**

Partially. While the paper discusses some limitations of the proposed approach, the discussion could be expanded to include potential boundary cases where the spectral assumption may not hold (e.g., low-frequency perturbations, color manipulations, or localized patch attacks). A brief discussion of these scenarios would help clarify the limitations of the proposed defense and its applicability under different threat models.

**Strengths And Weaknesses:**

**Strengths**:

1. **Simple yet effective defense mechanism**: The proposed approach is conceptually simple and effective. Instead of introducing heavy architectural modifications or large generative purification models, CSR leverages a fundamental spectral property of CLIP representations. The use of low-pass filtered features as a proxy for the clean manifold provides a practical and efficient way to guide feature rectification at test time.

2. **Training-free and broadly applicable at inference time**: A notable advantage of the method is that it operates entirely during inference without requiring model retraining. This property makes the approach particularly attractive for large pretrained models such as CLIP, where retraining is often computationally expensive or infeasible.

3. **Strong empirical performance**: The experimental evaluation is extensive and covers multiple vision tasks and datasets. CSR demonstrates substantial robustness improvements under strong attacks, including AutoAttack. In particular, the reported improvements over prior test-time defense methods such as TTC are significant and consistent across datasets and tasks.

4. **Insightful analysis of spectral vulnerability**: The paper provides a useful analysis of the spectral behavior of adversarial examples in CLIP representations. The identification of spectral fragility and its exploitation for defense is a valuable insight that may inspire further research on robustness in multimodal models.

**Weaknesses**:
1. **Limited discussion of adaptive attacks**: The proposed CSR recovery algorithm (Algorithm 1) is composed of differentiable operations. Therefore, a defense-aware adversary could potentially optimize perturbations that explicitly bypass the CSR objective L_{rec} and the gating condition C(x) \geq \tau.
Although the paper provides a theoretical explanation based on gradient conflict to argue that low-frequency attacks are difficult to optimize, empirical evaluation under fully adaptive attack settings would further strengthen the robustness claims.

2. **Sensitivity to similarity threshold (τ)**: The adversarial detection mechanism relies on a cosine similarity threshold \tau between the input image and its low-pass filtered version. The paper demonstrates strong detection performance (AUC > 0.95) across multiple datasets, which supports the effectiveness of the approach.
However, the robustness of the overall system with respect to the choice of \tau is not extensively analyzed. Since the filter radius r is fixed and the gating decision directly affects the rectification process, additional sensitivity analysis on the threshold parameter would help clarify the stability of the method across different domains.

3. **Reliance on the spectral assumption**: CSR relies on the assumption that adversarial perturbations primarily reside in mid-high frequency components, which motivates the use of low-pass filtered images as semantic anchors.
However, certain attacks—such as localized patch attacks or global color manipulations—may preserve their deceptive signals even after low-pass filtering. Since CLIP representations are known to be sensitive to color cues, such perturbations may not significantly alter the low-pass representation while still affecting the model’s semantic prediction. In such cases, the anchor itself may become contaminated, potentially guiding the rectification process in an incorrect direction. Additional discussion or evaluation of such boundary cases would help clarify the robustness of the spectral assumption underlying CSR.

---

> ### Author Rebuttal · Authors · 2026-03-30
>
> We sincerely thank the reviewer for recognizing CSR as an **insightful, effective, and broadly applicable** test-time defense framework. Your concerns mainly center on CSR’s capability boundaries (Q1 and Q3) and parameter ablations (Q2). We clarify and address the concern as follows.
>
> ---
>
> > Q1: Discussion of adaptive attacks ... empirical evaluation under fully adaptive attack settings would further strengthen the robustness claims.
>
> Thank you for the valuable question. Due to space constraints, we refer the reviewer to our response to Reviewer pvFZ (Q1) for detailed results under fully adaptive attack.
>
> In brief, although CSR shows some degradation in this setting, **it remains effective** and consistently outperforms TTC [CVPR'25]. We believe this robustness stems from CSR’s exploitation of the intrinsic spectral characteristics of the visual encoder, as discussed in Sec. 3.2.
>
> ---
>
> > Q2: Sensitivity to similarity threshold ($\tau$) ... additional sensitivity analysis on the threshold parameter would help clarify the stability of the method across different domains.
>
> Thank you for pointing this out. Empirically, **CSR remains relatively stable across the tested threshold interval of $\tau$**. Specifically, we provide evidence from the following two aspects.
>
> (1) In the main paper, the ablation study in Fig. 5 already shows that CSR maintains strong defensive performance under different radius-threshold ($r$, $\tau$) settings.
>
> (2) To further verify this point, we fix $r$ and vary $\tau$ from 0.78 to 0.86. As shown in Tables I, the classification accuracy remains relatively stable. We attribute this stability to the clear gap in feature consistency between benign and adversarial inputs, as illustrated in Fig. 3 of the main paper.
>
> **Table I**. Classification accuracy (%) of CSR under different similarity thresholds $\tau$.
>
> |              | $\tau$ = 0.78 | $\tau$=0.80 | $\tau$ = 0.82 | $\tau$ = 0.84 | $\tau$ = 0.86 |
> | ------------ | ------------- | ----------- | ------------- | ------------- | ------------- |
> | Clean        | 63.7          | 63.7        | 63.7          | 63.1          | 62.5          |
> | APGD (2/255) | 64.1          | 64.6        | 65.0          | 65.1          | 65.2          |
>
> *Regarding why APGD accuracy is slightly higher than clean accuracy, we note that, consistent with the observation in [1], adversarial examples may carry directional priors about the true decision boundary, which CSR can leverage to improve robustness.*
>
> [1] Liu, L. et al. "Adversarial Attacks Already Tell the Answer: Directional Bias-Guided Test-time Defense for Vision-Language Models." ICLR, 2026.
>
> ---
>
> > Q3: Reliance on the spectral assumption ... A brief discussion of these scenarios would help clarify the limitations of the proposed defense and its applicability under different threat models.
>
> Thank you for the constructive suggestion. To examine the boundary cases of CSR, we additionally evaluate two attacks beyond the standard $\ell_\infty$-bounded additive perturbation setting.
>
> (1) A targeted localized patch attack (strictly following [2]) with a 32$\times$32 corner patch, 100 optimization steps, learning rate 0.05, and a randomly sampled non-source target class.
>
> (2) A global color attack (strictly following [3]) that optimizes per-channel K-piece piecewise color curves in explicit color filter space for 50 steps (K=64, param_epsilon=8, step_size=0.5, kappa=0.0).
>
> Table II shows that **CSR remains effective under localized patch attacks**, achieving 57.3% accuracy, compared with 30.0% for TTC. Since patch attacks only affect a small spatial region, they do not substantially corrupt the global low-frequency semantic structure. Low-pass filtering further suppresses the patch influence, allowing CSR to retain a reliable semantic anchor for rectification. Under global color attacks, all methods provide a smaller improvement,  indicating that global color transformations are a less typical and more challenging boundary case for test-time defense, worthy of further study in future work. We will add a clearer discussion in the revised paper to explicitly discuss this boundary case and **clarify the corresponding scope** of applicability of CSR.
>
> **Table II**. Classification accuracy (%) under two boundary-case attacks.
>
> | Type          | Clean    | Localized patch attacks [2] | Global color attacks [3] |
> | ------------- | -------- | --------------------------- | ------------------------ |
> | Standard CLIP | **63.9** | 0.0                         | 2.2                      |
> | TTC [CVPR'25] | 40.9     | 30.0                        | **6.4**                  |
> | CSR (ours)    | 62.5     | **57.3**                    | 3.7                      |
>
> [2] Karmon et al., "*LaVAN: Localized and Visible Adversarial Noise*", ICML 2018.
>
> [3] Zhao et al., "*Adversarial Image Color Transformations in Explicit Color Filter Space*", TIFS 2023.

---

> > ### Author Rebuttal · Reviewer_kfam · 2026-04-03
> >
> > I thank for the authors for their comprehensive response and the additional experimental results provided during the rebuttal.
> >
> > The new analyses on fully adaptive attacks and threshold sensitivity effectively address my primary concerns regarding the robustness and stability of the proposed mechanism. I also appreciate the authors' effort in evaluating boundary cases such as localized patch and global color attacks, and their commitment to including these discussions in the revised manuscript.
> >
> > At this stage, I have no further technical concerns and will maintain my current rating. I look forward to making a final decision after reviewing the comments and discussions from the other reviewers.

---

> > > ### Author Response · Authors · 2026-04-04
> > >
> > > We sincerely thank the reviewer for the positive evaluation and constructive feedback. We are glad that our clarifications and additional results helped address the concerns, and we will include a discussion of this boundary case in the revised manuscript.

---

### Official Review · Reviewer_pvFZ · 2026-03-18

**Soundness:** 3
**Presentation:** 3
**Significance:** 3
**Originality:** 3
**Overall Recommendation:** 5
**Confidence:** 4

**Summary:**

This paper proposes Contrastive Spectral Rectification (CSR).as a new test-time defense against adversarial attack to resolve the exccessive computational cost and performance drops brought by adversarial fine-tuning (AFT). CSR utilizes the frequency dependence of adversarial examples and form a triplet structure to push adversarial examples back to the benign ones. Extensive experiments are conducted to evaluate the effectiveness of CSR.

**Compliance With Llm Reviewing Policy:**

Affirmed.

**Final Justification:**

My concerns haven been addressed by the authors, after considering reviews from other reviewers, I've raised my rating to 5.

**Key Questions For Authors:**

See weakness.

**Limitations:**

While the effectiveness of CSR is impressive regarding both defense and efficiency, the method has been contained within CLIP models and not evaluated on other VLMs. I do not see this as a major drawback or undermine the significance of the work, but this does have a negative impact on the wider application of CSR on larger VLMs such as Qwen, LLaVA, OpenFlamingo, etc.

**Strengths And Weaknesses:**

**Strength**

1. Challenges and motivations are clearly defined.

2. Extensive analysis and qualitative validation are provided for understanding the core idea of CSR.

3. Experiments are generally extensive and comprehensive covering several tasks and datasets.

**Weakness**:

I. **Adaptive attack**: Since CSR heavily relies on the intrinsic frequency traits of adversarial examples as its discriminative factor, which is not new in the field of adversarial attacks and could be kown by skilled attackers, it is neccesary to evaluate the robustness of CSR under the adaptive setting, i.e., attackers intentionally regularize high-freq reliance of perturbation optimziation. It would be more informative and comprehensive if the authors show how CSR behave under adaptive attacks.

II.**Layout**: 1.The layout of some of the tables, e.g., Table 3, can be improved and hinders reading currently.

2. There are a lot of pre-method results, experiements, and analysis, while the real method is relatively simple. I recommend using a more condensed version of all preimilaries so that the paper could be strcutured more tightly.

III.**Minor**: Table.1 and 2 are incorrectly hyperlinked to one of the references.

---

> ### Author Rebuttal · Authors · 2026-03-30
>
> We sincerely thank the reviewer for the **positive comments regarding our motivation and analysis**, and for highlighting the **effectiveness and efficiency** of CSR.
>
> We provide detailed responses to the concerns below.
>
> ---
>
> > Q1: Adaptive attack:  ... it is neccesary to evaluate the robustness of CSR under the adaptive setting, i.e., attackers intentionally regularize high-freq reliance of perturbation optimziation.  It would be more informative and comprehensive if the authors show how CSR behave under adaptive attacks.
>
> Thank you for the valuable suggestion. To address this concern, we conducted an **adaptive-attack** evaluation in which the attacker is assumed to **have full knowledge of CSR**. Specifically, we consider a strong adaptive attack based on APGD with an $\ell_\infty$ budget of 2/255. The attack optimizes the standard cross-entropy objective while additionally imposing a penalty on the mid-to-high frequency energy of the perturbation with the frequency radius set to 25 and the regularization weight set to 20.
>
> These results (Table I) lead to two main observations.
>
> **(1) The adaptive attack itself becomes less effective.**
>
> Despite using a larger perturbation budget (2/255), Standard CLIP still retains meaningful accuracy under the adaptive attack (e.g., 16.8% on Flowers102), whereas standard APGD (1/255) typically drives the accuracy close to 0.0%. This is consistent with Sec. 3.2: effective adversarial perturbations favor mid-to-high frequencies, and restricting them makes optimization much harder.
>
> **(2) CSR remains robust even when the attacker is fully aware of the defense.**
>
> Although CSR’s classification accuracy decreases under the adaptive attack, CSR still consistently outperforms TTC across all datasets. These results suggest that CSR remains effective even under defense-aware adaptation, confirming that its advantage persists even in this challenging setting.
>
> **Table I**. Classification accuracy (%) of CLIP-B/16 under adaptive attack (2/255).
>
> | Method        | ImageNet | Flowers102 | Sun397   |
> | ------------- | -------- | ---------- | -------- |
> | Standard CLIP | 14.1     | 16.8       | 12.5     |
> | TTC [CVPR'25] | 29.1     | 35.3       | 33.5     |
> | CSR (ours)    | **46.3** | **42.6**   | **51.7** |
>
> ---
>
> > Q2: Layout: The layout of some of the tables, e.g., Table 3, can be improved and hinders reading currently ... I recommend using a more condensed version of all preimilaries.
>
> Thank you for this helpful suggestion. In the revision, we will adjust the table placement to improve readability and visual balance, specifically by moving Tables 3 and 4 to Page 7 and arranging them side by side, while relocating Tables 6 and 7 to Page 8. We will also condense the preliminaries so that the paper is structured more tightly.
>
> We believe these **layout adjustments** will make the presentation more polished.
>
> ---
>
> > Q3: Minor: Table.1 and 2 are incorrectly hyperlinked to one of the references.
>
> We will fix the incorrect hyperlinks for Tables 1 and 2 in the revision. Thank you for pointing this out.
>
> ---
>
> > Q4: While the effectiveness of CSR is impressive regarding both defense and efficiency, the method has been contained within CLIP models and not evaluated on other VLMs. I do not see this as a major drawback or undermine the significance of the work, but this does have a negative impact on the wider application of CSR on larger VLMs such as Qwen, LLaVA, OpenFlamingo, etc.
>
> Thank you for the thoughtful comment. We would like to clarify that **CSR extends beyond CLIP and remains effective on larger VLMs**. We support this point with the following evidence.
>
> (1) In the main paper, we already evaluate CSR on **LLaVA-1.5** for both VQA and captioning tasks. Table 7 reports a 43.0% robustness gain, Figure 6 provides qualitative examples, and Appendix E.3 describes the implementation details.
>
> (2) We further evaluate CSR on **Qwen2.5-VL**  under the standard M-Attack [1] protocol with a perturbation budget of 16/255. As shown in Table II, on adversarial examples for the captioning task, CSR improves the accuracy from 18% to 57%, substantially outperforming TTC.
>
> **Table II**. Results on Qwen2.5-VL under the standard M-Attack [1] protocol. We report captioning accuracy (%) and average similarity to the ground-truth answer.
>
> | Method                     | Accuracy (↑) | AvgSim. (↑) |
> | -------------------------- | ------------ | ----------- |
> | Qwen2.5-VL                 | 18.0         | 0.149       |
> | Qwen2.5-VL + TTC [CVPR'25] | 21.0         | 0.154       |
> | Qwen2.5-VL + CSR (ours)    | **57.0**     | **0.381**   |
>
> [1]  Li et al. "A Frustratingly Simple Yet Highly Effective Attack Baseline: Over 90% Success  Rate Against the Strong Black-box Models of GPT-4.5/4o/o1". NeurIPS 2025.

---

> > ### Author Rebuttal · Reviewer_pvFZ · 2026-04-04
> >
> > I appreciate the authors' thorough response for my concerns and questions. I have no more questions and will finalize my score after a joint consideration with review from other reviewers.

---

> > > ### Author Response · Authors · 2026-04-04
> > >
> > > We sincerely appreciate the reviewer’s positive assessment and thoughtful consideration. We are glad that our responses addressed the concerns. These comments will help further polish and strengthen our work.

---

### Decision · Program_Chairs · 2026-04-30

**Decision:**

Accept (regular)

**Comment:**

In this work, the authors propose an efficient test-time defense named Contrastive Spectral Rectification (CSR) to address the limitations of Vision-language models (VLMs) in resisting adversarial examples (AEs).
After rebuttal, three of the four reviewers have been satisfied with authors' responses and recognized the contributions of this paper, including (1) Extensive analysis and qualitative validation are provided for understanding the core idea of CSR; (2) Insightful analysis of spectral vulnerability; (3) Training-free and broadly applicable at inference time; and (4) Novel and sound approach.
Although Reviewer e59U still has concerns regarding "a spectral fragility in adversarial examples," "sensitivity to mid-to-high frequency components," and the rectification budget, it seems to the AC that these concerns have been properly addressed.
Overall, this work exhibits several merits ad outweighs the possible concerns. Thus, the AC recommends this paper be accepted but the authors is requested to address the remaining concerns clearly in the final version!